# The Basics of Covalent Bonding in Terms of Energy and Dynamics

**DOI:** 10.3390/molecules25112667

**Published:** 2020-06-08

**Authors:** Sture Nordholm, George B. Bacskay

**Affiliations:** 1Department of Chemistry and Molecular Biology, The University of Gothenburg, SE-412 96 Göteborg, Sweden; 2School of Chemistry, The University of Sydney, Sydney, NSW 2006, Australia

**Keywords:** covalent bond, kinetic energy, delocalization, interatomic electron motion

## Abstract

We address the paradoxical fact that the concept of a covalent bond, a cornerstone of chemistry which is well resolved computationally by the methods of quantum chemistry, is still the subject of debate, disagreement, and ignorance with respect to its physical origin. Our aim here is to unify two seemingly different explanations: one in terms of energy, the other dynamics. We summarize the mechanistic bonding models and the debate over the last 100 years, with specific applications to the simplest molecules: H_2_^+^ and H_2_. In particular, we focus on the bonding analysis of Hellmann (1933) that was brought into modern form by Ruedenberg (from 1962 on). We and many others have helped verify the validity of the Hellmann–Ruedenberg proposal that a decrease in kinetic energy associated with interatomic delocalization of electron motion is the key to covalent bonding but contrary views, confusion or lack of understanding still abound. In order to resolve this impasse we show that quantum mechanics affords us a complementary dynamical perspective on the bonding mechanism, which agrees with that of Hellmann and Ruedenberg, while providing a direct and unifying view of atomic reactivity, molecule formation and the basic role of the kinetic energy, as well as the important but secondary role of electrostatics, in covalent bonding.

## 1. Introduction

Covalent chemical bonding is undoubtedly a central concept in Chemistry. While bond formation is arguably the most fundamental chemical process, its physical origin is still the subject of debate, even today when accurate quantitative molecular electronic structure calculations of ever-increasing accuracy and complexity have become widely available. Seemingly, there is a chasm between numerical and physical resolutions of the covalent bond. It is our aim to improve the physical understanding of bonding and help connect the physical and numerical views of bonding by drawing on the duality between energy and time present in quantum mechanics.

The idea of shared electron pairs corresponding to chemical bonds was introduced by Lewis [1] over a hundred years ago in a landmark publication, a decade before Schrödinger [2] developed the method that effectively laid the foundations of quantum chemistry and provided the tools with which Lewis’ ideas could be rigorously tested and interpreted. Lewis’ model built on the earlier work of Abegg [3] who had introduced and explored the Octet Rule. Abegg’s work was also inspirational in Kossel’s development of ionic bonding [4]. The theories of Abegg, Lewis, and Kossel were extended by Langmuir [5,6,7,8] who extended the Octet Rule by developing the 18- and 32-electron rules and introduced the name “Covalent Bond” for a shared pair of electrons. The Lewis–Kossel–Langmuir theory of covalent bonding is considered a basic tenet of Chemistry and is widely taught in senior high school as well as university chemistry courses.

The development of quantum theory provided a theoretical underpinning of covalent bonding, via Burrau’s [9,10] quantum mechanical calculation on the hydrogen molecule ion H_2_^+^ and, in the same year, the Heitler–London calculation [11] of the geometry and bond energy of the H_2_ molecule. Burrau’s work [9,10], in good agreement with experiment [12,13,14], provided theoretical support for the existence of one electron covalent bonds. The Heitler–London work [11] especially made the connection between the Lewis shared pair of electrons [1] and its physical quantitative description. The valence-bond (VB) theory of Pauling [15,16] has provided a natural bridge between these two theories and, especially in its qualitative form, for many years it has been the bonding theory of choice of most chemists. Molecular orbital (MO) theory, promoted initially by Mulliken [17], Hund [18], Hückel [19], and others [20,21,22,23,24,25] has, however, come to be preferred by the chemical community, especially when dealing with multicenter bonds, and more generally for problems of electronic excitation, reactivity, and transition metal chemistry. By the 1960s and 70s, when the development and distribution of quantum chemical computer codes had become widespread, the majority of the methods, such as those in the Gaussian suite of programs [26,27], first released in 1970, were Hartree–Fock Self-Consistent Field MO (HF-SCF) [28,29] based. Recently, however, the VB method has seen a renaissance and has been demonstrated to be the natural approach to a number of important chemical problems [30,31,32,33].

With respect to the basic physics of covalent bonding a long-held and still widespread view is that chemical bonding is essentially an electrostatic phenomenon. Namely, the energy lowering that corresponds to bond formation is thought to be the result of the decrease in potential energy due to the attractive interaction between the nuclei and the electronic charge that is accumulated in the bond region. This essentially classical and static picture of interacting charge distributions is appealing in its simplicity, indeed it appears to be a straightforward extension of the Lewis theory. Moreover, the above electrostatic view appears to be consistent with the Virial Theorem [34,35,36], according to which the ratio of total potential to kinetic energy of a molecule at its equilibrium geometry, always equals −2. In other words, the attractive component of the binding energy is therefore due to the (electrostatic) potential energy, whereas the kinetic component is repulsive. This electrostatic view was originally advanced by Slater [36] in 1933, supported by Feynman [37] in 1939, and later by Coulson [24], whose book *Valence* of 1952 has had a strong influence on the chemical community. More recently, Bader [38,39] has also expressed strong support of Slater’s model of covalent bonding.

In contrast to the electrostatic view, Hellmann [40] proposed that covalent bonding should be understood as a quantum mechanical effect, brought about by the lowering of the ground state kinetic energy associated with the delocalization of the motions of valence electrons between atoms in a molecule. For several decades this kinetic view [40] was ignored by most chemists, possibly because it went against the already accepted seemingly simpler electrostatic explanation [36], and because Hellmann had based his reasoning on the statistical Thomas−Fermi model [41,42], that was subsequently shown to be unable to describe covalent bonding [43,44,45]. Other contributing factors could have been the apparent conflict with the Virial Theorem [34,35,36] that Hellmann was unable to resolve [46] and, quite likely, his early tragic death. Interestingly though, the physicists Peierls [47] and Platt [48] expressed general agreement with Hellmann’s view, as early as 1955 and 1961, respectively. We note, however, that the above problem, implicit in the Thomas–Fermi model [43,44,45], became our own point of entry into the dynamic analysis of bonding in 1987, 50 years after Hellmann’s work [46].

The contradiction between these two different qualitative models of the covalent bonding mechanism required a rigorous in-depth analysis for its resolution. The analysis was carried through, on the basis of the quantum mechanical Variation Principle, by Ruedenberg and coworkers [49,50,51,52,53,54,55,56,57,58,59] from 1962 on, first for H_2_^+^ and H_2_ and later for other homonuclear diatomic molecules as well. These investigations showed that covalent bonding is a quantum effect as originally suggested by Hellmann [40], since the critical component of bonding is interatomic electron delocalization, which is the quantum mechanical term for electron sharing. Stabilizing electron delocalization in ground states is associated with combination and constructive interference of the atomic orbitals (AOs) of the atoms in a molecule to form molecular orbitals (MOs). This normally results in bonding by a net decrease in the kinetic energy of the molecule, in agreement with Hellmann’s view [40]. However, as the internuclear distance becomes smaller than about twice the equilibrium separation, an additional, more complex, effect comes into play, namely intra-atomic (orbital) contraction. In a minimal atomic basis set, as the term implies, orbital contraction corresponds to an increase in the orbital exponents of the basis functions, resulting in tighter orbitals, and thus electron densities, around the nuclei. In practice, the orbital exponents, being non-linear parameters, are optimized at each distinct geometry, by numerical minimization of the molecular energy. (Alternatively, the use of extended but geometry independent basis sets, such as those in state-of-the-art HF-SCF or density functional calculations, allows orbital contractions to be resolved via the optimization of the occupied MOs). The net result of orbital contraction is additional stabilization by a decrease in the potential energy while the kinetic energy is increased. These energy shifts ensure that the Virial Theorem [34,35,36] is satisfied. As discussed elsewhere, the orbital contractions actually enhance the degree of interatomic delocalization and thereby lower the interatomic kinetic energy further, even though they increase the antibonding intra-atomic deformation energies. The net result is a decrease in the total energy, as demanded by the Variation Principle. The Hellmann–Ruedenberg view of covalent bonding has been accepted and adopted by many theoretical chemists [60,61,62,63,64,65,66,67,68,69,70,71,72,73,74,75,76,77,78,79,80,81,82,83,84,85,86,87], including Fukui [62] and Mulliken [63].

Interestingly, after expressing support for Slater’s electrostatic view of covalent bonding in 1939, 26 years later Feynman [88] himself, in his famous *Lectures on Physics*, explained covalent bond formation in H_2_^+^, and by extension in other molecules, as the consequence of a flip-flop motion of electrons between bonded atoms, causing a corresponding drop in the electron’s kinetic energy, as a molecule forms. His simplest argument was based on the Heisenberg uncertainty principle, Δ*x*Δ*p* ≥ *ħ*/2, that would imply that the lower energy of the electron in the bonding stationary state of the molecule is a consequence of delocalization, i.e., “spreading out” or increasing Δ*x*, that results in a drop in kinetic energy (proportional to (Δ*p*)^2^) without a significant increase in its potential energy. The opposite would hold for the repulsive antibonding state. This conclusion is in complete agreement with the views of Hellmann [40] and Ruedenberg [49]. Feynman [88] was, so far as we know, first to emphasize the close connection between electron dynamics and kinetic energy in covalent bonding, i.e., between the flip-flop motion and the corresponding kinetic energy lowering. Unfortunately, his clear sighted reasoning, if after a change of heart, seems to have largely by-passed the attention of the chemical community.

While our own work over a period of 25 years [68,69,72,75,77,78,79,80,81,82,83], has agreed with Hellmann’s [40], Ruedenberg’s [49,50,51,52,53,54,55,56,57,58,59], and Feynman’s [88] views, it has also expanded on them by exploring the quantum dynamical description of covalent bonding. Noting that Thomas–Fermi (TF) theory [41,42], the original and simplest type of density functional theory (DFT) [45,89], is unable to describe covalent bonding [43,44,45], we analyzed the reasons for this failure in an effort to better understand the basic physics of bonding [68,78,79,80]. We have found that, at least in simple systems, the source of the problem is the simplified semi-classical form of the TF kinetic energy functional, resulting in a theory that is unable to account for dynamical constraints (non-ergodicity) and slow electron transfer and thus for any hindered internal electron dynamics in atoms and molecules. As it is the relaxation of these dynamical constraints that will facilitate interatomic electron transfer in molecules, i.e., electron sharing, the natural conclusion is that covalent bonding is a dynamical process which is implicit in the phenomenon of delocalization [80].

The first suggestion that covalent bonding was best seen as a quantum dynamical phenomenon was, as noted above, published by Feynman [88] in 1965, in his wide ranging lectures on physics, three years after Ruedenberg’s groundbreaking paper in Reviews of Modern Physics [49]. Feynman [88], unlikely to have read Ruedenberg’s paper, treats H_2_^+^ as a simple two-state system of bonding and antibonding MOs and uses time-dependent quantum theory to show that the bonding electron in H_2_^+^, must, if localized, oscillate between the two nuclei with a frequency that is directly related to the energy difference between the delocalized MOs. These MOs are stationary states and their energy difference is approximately twice the bond energy. As noted above, this energy difference, part of the time-independent quantum description of the molecule and due to the orthogonality of ground and first excited states, is essentially kinetic in character. However, at the time-dependent level, the degree of bonding could be seen to be proportional to the rate of interatomic electron oscillation, which in turn is dependent on delocalization of the electron. The dynamical picture of interatomic electron oscillation Feynman [88] referred to as the “flip-flop mechanism” of covalent bonding. He recognized that the mechanism could as well be seen in terms of wave function delocalization combined with energy splitting between molecular orbitals formed from localized atomic orbitals.

Ruedenberg and co-workers [49,50,51,52,53,54,55,56,57,58,59,83] reached an equivalent energy analysis of bonding utilizing the results of time-independent variational calculations, their main conclusion being that the physical basis of bonding is interatomic electron delocalization. They went on to analyze the importance of kinetic energy to bonding, the role of orbital contraction and the Virial Theorem [34,35,36]. The latter concept and theorem were actually ignored by Feynman [88], since they do not immediately arise in the dynamical approach to bonding.

In addition to his treatment of H_2_^+^, Feynman [88] discussed bonding in H_2_ as well as in benzene and (conjugated) dye molecules, as other examples of two-state systems. He demonstrated the generality of his idea of attraction by particle exchange between attractive centers by the analogous treatment of the umbrella inversion in ammonia and the interaction of nucleons. The covalent bonding in molecules was found to be an example of a quite general phenomenon in physics.

As noted above, the quantum dynamical view of covalent bonding is not unique. Nor is at odds with the Hellmann–Ruedenberg theory [40,49,50,51,52,53,54,55,56,57,58,59,83] or the Virial Theorem [34,35,36]. Instead it provides a fully consistent alternative interpretation which sheds light on covalent bonding while avoiding, or in a deeper analysis helping to resolve, the apparent contradiction between the theory and the theorem. The next section of this paper will review the salient features of Ruedenberg’s [49,50,51,52,53,54,55,56,57,58,59] theory and our contributions to it [68,69,72,75,77,78,79,80,81,82,83]. We regard it as most useful for those who want to understand covalent bonding in terms of time-independent interpretations of concepts such as electron density, delocalization and energy. The quantum dynamical mechanism [68,75,78,79,80,81] is provided by the duality of representations offered to us by quantum mechanics [88] such that we can choose to see the bonding mechanism in terms of energy as well as in terms of dynamics. We propose to employ both representations to show that this duality of views of bonding is advantageous since it makes clear: (i) that bonding is a quantum phenomenon relating to both energy and dynamics and (ii) how the rate of interatomic electron motion, i.e., delocalization and its timescale, is the key determinant of the bonding while related mechanisms of orbital contraction or electron correlation are important but secondary.

Much of the early work on the basic physics of chemical bonding was done in tandem with the developing new branch of science: Quantum Chemistry. Indeed the earliest quantum mechanical computations, ab initio and semi-empirical, focused on questions of bonding, but gradually the balance shifted and nowadays the majority of quantum chemical calculations, performed typically on supercomputers, focus on modeling chemical processes, structure, and properties. Yet, as chemists we want to, indeed need to, understand bonding and considerable effort is directed to the extraction of simple-to-understand bonding information from complex wave functions that are often characterized by millions of numbers [90,91,92,93,94,95,96,97,98,99,100,101,102,103,104]. In contrast, this paper is concerned with the fundamental aspects of bonding, a problem of long standing, rather than the immediate interpretation of results from large scale quantum chemical calculations.

We summarize the analysis of covalent bonding in H_2_^+^ and H_2_ within the energy picture in Section 2 below, ending with a discussion of the corresponding time-dependence of an electron initially confined to one of the protons in H_2_^+^. Thus we demonstrate the direct relation between the energy and the dynamical view of bonding in much the same terms as Feynman [88]. We then, in Section 3, discuss the general analysis of covalent bonding in the time-dependent picture and end with examples of the benefits of drawing on both pictures, time-dependent as well as time-independent, to achieve a deeper and more general understanding of the bonding mechanism. In Section 4, finally, we reflect on the circumstances that 100 years ago set many chemists on the path to an oversimplified view of bonding in terms of electrostatics that made us overly resistant to a correct identification of delocalization and corresponding easing of kinetic energy as the key to bonding. Hopefully, the combination of energetics and dynamics will provide an understanding of the covalent bond that’s both physically clear and fully consistent with the results of quantum chemistry.

## 2. Energy Analysis of One- and Two-Electron Bonds: The H_2_^+^ and H_2_ Molecules

These are the simplest prototypes of molecules with covalent bonds, involving just one electron in the H_2_^+^ molecule or the archetypal Lewis pair in H_2_. The simplest molecular wave functions are constructed from the (exact) atomic orbitals (AOs) of a hydrogen atom. The spatial components are:(1)$$\Psi (H_{2}^{+};\zeta ,R)={\left[2\left(1+S_{ab}\right)\right]}^{-1\;2}[\phi_{a}(\zeta )+\phi_{b}(\zeta )]\;,$$
and:(2)$$\Psi \left(H_{2};\zeta ,R\right)={\left[2\left(1+S_{ab}^{2}\right)\right]}^{-1\;2}\;\left[\phi_{a}(1)\phi_{b}(2)+\phi_{b}(1)\phi_{a}(2)\right]\;,$$
where the overlap integral *S_ab_*, defined as:(3)$$S_{ab}=\langle \phi_{a}|\phi_{b}\rangle =\int \phi_{a}\left(r\right)\phi_{b}\left(r\right)dr,$$
is dependent on the orbital exponent *ζ* and the internuclear separation *R*. (The full wave functions for the ground states are obtained by multiplying the above spatial functions by the doublet and singlet spin eigenfunctions for H_2_^+^ and H_2_, respectively). The H_2_ wave function of Equation (2), being the linear combination of atomic configurations, is the archetypal VB wave function [15,16], as originally proposed by Heitler and London [11]. The coordinates of the two electrons are simply written as 1 and 2. Because it smoothly dissociates to H atoms, in this work it is preferred to the MO wave function (i.e., a doubly occupied *σ_g_* MO) that predicts a mixture of H atoms and H^+^/H^−^ ions as *R* → ∞. We note, however, that in the case of H_2_^+^ the single electron wave function of Equation (1) could equally well be regarded as MO or VB type. The normalized AOs are just 1*s*-type AOs, e.g.,
(4)$$\phi_{a}={\left(\zeta^{3}/\pi \right)}^{1/2}\exp\left(-\zeta r_{a}\right)\;,$$
where *r_a_* is the distance from nucleus *a*. The optimized orbital exponents *ζ* for H_2_^+^ and H_2_ vary between the separated H atoms limit of 1.0 (*R* = ∞), and the united He^+^ and He atom limits of 2.0 and 1.688, respectively (*R* = 0), in atomic units (see Appendix A).

While these simplest of wave functions can be improved upon so that their predictions become quantitative, e.g., by the inclusion of polarization functions and in the case of H_2_ account for a greater degree of electron correlation, it has been found, more than once, that the basic physics of covalent bonding are adequately resolved by the above minimal sets [56,57,81,82]. Keeping the calculations as straightforward and simple as possible means that the essential elements of bonding can be clearly resolved with as little mathematical complexity as possible.

### 2.1. Bonding Energetics

With *ζ* optimized at each distance (full lines), as well as fixed at the H atom value of 1.0 (dashed lines), the computed energy curves of H_2_^+^ and H_2_ are shown in Figure 1. Optimization of the exponent results in a greater degree of bonding in both systems and shorter bond lengths. The computed equilibrium bond lengths and binding energies are summarized in Table 1.

The qualitative similarities between the H_2_^+^ and H_2_ energy curves are obvious. With the exponent *ζ* fixed at 1, the total (electrostatic) potential energies for both molecules are repulsive at all distances, a clear indication that bonding takes place because of the decrease in the kinetic energy. This is a quantum effect, i.e., a consequence of the quantum mechanical nature of the electron, in particular the behavior of its kinetic energy. Thus, a drop in kinetic energy is indicative of the electron being less constrained, i.e., having more room to move in. This is the conclusion that was reached by Hellmann [40] in 1933 and also by Feynman [88], as discussed in the third volume of his famous *Lectures on Physics* series, published in 1965. Feynman treats the H_2_^+^ and H_2_ molecules as examples of two-state systems, where the back and forth flip of the electron(s) (from one nucleus to the other), i.e., delocalization, a consequence of the quantum nature of electrons, produces the bonding in both systems.

Optimization of the orbital exponent *ζ* (resulting in increasingly larger values than 1.0 as *R* decreases), yields greater stability as well as shorter equilibrium bond lengths that are in close agreement with experiment. As in a H atom (where the kinetic and potential energies are *ζ*^2^/2 and −*ζ*, respectively), this process of *orbital contraction* in the molecules is accompanied by an increase in the kinetic energy but a greater degree of drop in the potential energy, so that the Virial Theorem [34,35,36,105,106,107], is satisfied precisely at the equilibrium geometry. (According to this theorem, for any system of charges at equilibrium, molecule or atoms, the ratio of potential (*V*) to kinetic energy (*T*) is exactly −2. Therefore, the same ratio holds for the potential (Δ*V*) and kinetic (Δ*T*) components of the binding energy Δ*E*). While the overall effect of orbital contraction is to strengthen the bond, the actual shifts in the total bond energies are minor and essentially intra-atomic in nature, so the key to the bonding can clearly be identified as the decrease in interatomic kinetic energy [58,59] (as demonstrated also in later sections of this paper). This, of course, is in direct contradiction to the electrostatic theory that claims that the drop in potential energy, as stipulated by the Virial Theorem, [34,35,36,105,106,107] is due to the electrostatic interaction of the increased electronic charge in the interatomic region with the nuclei [24,36,37,38,39].

The orbital contraction and its effects on the equilibrium geometries and energies, so as to satisfy the Virial Theorem [34,35,36,105,106,107], can be obtained, to a very good approximation, by a simple scaling procedure [105,106,107], that yields:(5)$$\zeta =-\frac{V\left(1\right)}{2T\left(1\right)}\;,\;R_{e}\left(\zeta \right)=\frac{R_{e}\left(1\right)}{\zeta }\;,\;E\left[R_{e}\left(\zeta \right)\right]=\zeta^{2}T\left(1\right)+\zeta V\left(1\right)\;.$$

For H_2_^+^, application of this procedure yields *ζ* = 1.238, thus, an equilibrium distance of 2.01 *a*_0_ and a total binding energy of −0.086 *E*_h_. In the case of H_2_, we obtain *ζ* = 1.166, i.e., an *R*_e_(*ζ*) of 1.41 *a*_0_ and a total binding energy of −0.139 *E*_h_. The agreement with the results of independent orbital optimization, as summarized in Table 1, is excellent for both systems. Thus, as demonstrated above for H_2_^+^ and H_2_, orbital optimization is essentially a rescaling of the molecular wave functions as well as their constituent AOs. In other words, the delocalization of electrons, hence bond formation, is effectively between contracted atoms, exactly as suggested by Ruedenberg [49] in his first publication on this topic in 1962.

### 2.2. Molecular Density and Delocalization

By way of further exploration of the phenomenon of covalent bonding, consider the change in electron density that takes place as a molecule, H_2_^+^ or H_2_, forms from the constituent atoms and/or nuclei. The molecular density $\rho \left(\zeta ,R\right)=n_{\mathrm{el}}\Psi^{2}(\zeta ,R)$ (at any internuclear separation *R* and 1*s* AOs with exponent *ζ* and where *n*_el_ is the number of electrons, i.e., 1 or 2 for H_2_^+^ or H_2_, respectively) is decomposed into quasi-classical, *ρ*_qc_, (atomic) and interference, *ρ*_I_, contributions, i.e.,
(6)$$\rho =\rho_{\mathrm{qc}}+\rho_{I}\;,$$
where:(7)$$\rho_{\mathrm{qc}}\left(\zeta \right)=\frac{n_{\mathrm{el}}}{2}\left[\phi_{a}^{2}\left(\zeta \right)+\phi_{b}^{2}\left(\zeta \right)\right]\;.$$

One-dimensional plots of the densities of H_2_^+^ as functions of the internuclear coordinate *z* as well as the resulting interference densities are shown in Figure 2. The bonding state’s wave function (Equation (1)) is an in-phase combination of the AOs *φ_a_* and *φ_b_* and their constructive interference results in a build-up of density in the bond region, i.e., in-between the nuclei, with a negative interference contribution close to the nuclei. The opposite holds for the antibonding state, i.e., negative interference in the bond region and an increased density around the nuclei. Note that the very existence of an antibonding state is a quantum effect, as it is the consequence of the quantum mechanical (wave) nature of the electrons. At *R* = 2.5 *a*_0_ its energy is 0.209 *E*_h_ above that of the separated atoms H + H^+^, i.e., it is a repulsive state, because of its large 0.316 *E*_h_ kinetic energy, despite an attractive −0.106 *E*_h_ potential energy contribution (both relative to H + H^+^). The large kinetic energy is a consequence of the node in the wave function, i.e., a region of large gradients (in an absolute sense), consistent with the quantum nature of the electron.

At any internuclear distance *R*, the interference of the AOs is related to their overlap integral, *S_ab_*, which plays a crucial role in the energy expressions of both H_2_^+^ and H_2_. In particular, the kinetic energies are:(8)$$T\left(H_{2}^{+}\right)=\frac{T_{aa}+T_{bb}+2T_{ab}}{2\left(1+S_{ab}\right)}=\frac{T_{aa}+T_{ab}}{1+S_{ab}}\;,$$
(9)$$T\left(H_{2}\right)=\frac{2T_{aa}+2T_{bb}+4T_{ab}S_{ab}}{2\left(1+S_{ab}^{2}\right)}=\frac{2\left(T_{aa}+T_{ab}S_{ab}\right)}{1+S_{ab}^{2}}\;,$$
where $T_{ab}=\langle \phi_{a}|\hat{T}|\phi_{b}\rangle$ and from symmetry $T_{aa}=T_{bb}$. According to our calculations, at distances larger than ~ 3 *a*_0_ the diagonal term *T_aa_* is significantly larger in magnitude than the off-diagonal *T_ab_*, indicating that the dominant contribution to the total kinetic energy in the case of H_2_^+^ comes from the quotient *T_aa_*/(1+*S_ab_*), as shown by the plots in Figure 3. Thus, at bond lengths larger than ~ 3 *a*_0_, i.e., quite near the equilibrium distance of 2.5 *a*_0_ (when the orbital exponent is fixed at the atomic value of 1.0 and the kinetic energy is effectively at its minimum) the contribution of the off-diagonal kinetic coupling term *T_ab_* is essentially negligible. Binding, i.e., the drop in kinetic energy, is effectively due to the rapid increase in the overlap *S_ab_*. At smaller distances (in the region *R* ≤ ~ 3 *a*_0_) *T_ab_* is no longer negligible, indeed it is responsible for the repulsive kinetic energy contribution. The same conclusions apply in the case of H_2_, except the critical distance (where the repulsive effect of *T_ab_* becomes non-negligible) is smaller, ~ 2.5 *a*_0_. The effect of orbital contraction, i.e., optimization of the exponent *ζ*, noticeable at distances smaller than ~ 4.5 *a*_0_ in the case of H_2_^+^, is to increase the magnitude of *T_aa_* (since *T_aa_* = *ζ*^2^/2) that results in the observed increase in total kinetic energy.

Orbital contraction, as the data in Figure 1 and Table 1 indicate, results in further stabilization of the molecules as well as ensuring that the Virial Theorem [34,35,36,105,106,107] is satisfied. As shown, however, in Figure 4, full geometry and exponent optimization results in higher interference density (difference between densities of molecule and *contracted* or quasi-atoms) in the bond region than obtained with *ζ* = 1, as well as a correspondingly larger decrease in the density near the nuclei. Relative to uncontracted H atoms (*ζ* = 1) the effect of orbital contraction in the molecule is a prominent increase in the density around the nuclei but also, to a lesser extent, in the bond region. As discussed above, the effect is an overall contraction of the molecular wave function, hence density.

The physical origin of orbital contraction, in particular why it is limited to shorter internuclear distances, is elucidated in detail in previous publications [58,59,83]. Briefly, the additional interference density that results from orbital contraction brings about an additional interatomic kinetic energy lowering, although at the expense of the intra-atomic contribution. Overall, however, the process brings about a lowering in the total energy.

That a build-up of electron density in the bond region occurs has been long known, especially as its existence can be illustrated via qualitative arguments or via low level hand calculations. Unfortunately, it has been too tempting to conclude, incorrectly, that the energetic explanation of bonding must be due to the extra charge in the bond region, i.e., its electrostatic attraction to the nuclei.

A more complete illustration of the magnitude and distribution of the interference density in H_2_^+^ is provided by the contour map in Figure 5, which clearly shows the movement of charge relative to the quasi (contracted) atoms as well as the sheet of zero *ρ**_I_* that separates the regions of density buildup and loss. The total charge *Q*_I_ that is actually moved into the interatomic (bonding) region can be computed by numerical integration [82,83] (although in the case of H_2_^+^ an analytic expression has been derived [50] and subsequently also used by Schmidt et al. [58,59]). Using the numerical information on the location of the zero *ρ**_I_* sheet, it is also possible to compute the kinetic and nuclear attraction energies associated with the charge movement, i.e., their values in the regions of positive and negative *ρ**_I_*.

In summary, the buildup of electron density in the internuclear region beyond the quasi-classical sum of the atomic densities, a quantum mechanical consequence of the constructive interference of electron waves that are specified in terms of AOs, is a process that accompanies the formation of a covalent bond. The buildup of density in the bond is not caused by, nor does it result in, a drop in potential energy. Constructive interference is a precondition of electron delocalization, i.e., interatomic electron flow, which results in a decrease in kinetic energy. An increase in the orbital exponent *ζ* on the one hand results in a tighter electron density around the nuclei but also an increase in the interference density in the bond region. Ultimately, it also leads to a shorter and stronger bond. While electron delocalization leads to a drop in the (quantum mechanical) kinetic energy of the electron as well as increased electrostatic attraction of the electrons to the nuclei (i.e., a lower potential energy), an increase of the orbital exponent results in tighter, mostly atomic, contribution to the density. The latter brings, as envisaged by Feinberg et al. [51], an increase in the *nuclear suction*, as well as in the *kinetic pressure*, i.e., a larger (more negative) potential energy of attraction and a higher kinetic energy.

### 2.3. Intra- and Interatomic Contributions to Bonding Energies

An obvious advantage of constructing molecular wave functions Ψ in terms of AOs, i.e., *φ*_a_ and *φ*_b_, is that the molecular electron density and energies are readily decomposable into intra- and interatomic contributions. Hence, the total molecular energy relative to its dissociation products is simply [58,59]:(10)$$\Delta E=E_{\mathrm{intra}}+E_{\mathrm{inter}}\;=T_{\mathrm{intra}}+V_{\mathrm{intra}}+T_{\mathrm{inter}}+V_{\mathrm{inter}}\;.$$

In the current case of H_2_^+^ and H_2_, the intra-atomic kinetic and potential energies due to contraction are defined as:(11)$$T_{\mathrm{intra}}=n_{\mathrm{el}}\left(\zeta^{2}-1\right)/2\;\;,\;V_{\mathrm{intra}}=-n_{\mathrm{el}}\left(\zeta -1\right)\;,\;E_{\mathrm{intra}}=T_{\mathrm{intra}}+V_{\mathrm{intra}}\;,$$
since the energies of a H atom with wave function *φ*(*ζ*) are:(12)$$T\left(H,\zeta \right)=\zeta^{2}/2\;,\;V\left(H,\zeta \right)=-\zeta \;.$$

The interatomic components of the various energies are then the molecular kinetic, potential, and total energies relative to the contracted atoms, i.e.,
(13)$$T_{\mathrm{inter}}\left(\zeta \right)=\langle \Psi \left(\zeta \right)|\hat{T}|\Psi \left(\zeta \right)\rangle \;-n_{\mathrm{el}}T\left(H,\zeta \right)\;,$$
(14)$$V_{\mathrm{inter}}\left(\zeta \right)=\left[\langle \Psi \left(\zeta \right)|{\hat{V}}_{a}+{\hat{V}}_{b}+\left(n_{el}-1\right){\hat{V}}_{12}\;|\Psi \left(\zeta \right)\rangle +V_{N}\right]-n_{\mathrm{el}}V\left(H,\zeta \right)\;,$$
(15)$$E_{\mathrm{inter}}\left(\zeta \right)=T_{\mathrm{inter}}\left(\zeta \right)+V_{\mathrm{inter}}\left(\zeta \right)\;,$$
where $\hat{T}$ is the kinetic energy operator, *V*_N_ is the nuclear repulsion energy, ${\hat{V}}_{12}$ is the inter-electron repulsion operator in H_2_ and ${\hat{V}}_{a}$ and ${\hat{V}}_{b}$ are the potential (nuclear attraction) energy operators. The interference component of the kinetic energy, *T*_I_, is actually the same as the interatomic term, *T*_inter_. Clearly, increasing the orbital exponent results in a tighter AO, i.e., an increase in its kinetic energy and a corresponding decrease in its potential energy of attraction to the nuclei as summarized by the above Equations (11) and (12).

The interatomic potential energy contribution for H_2_^+^ can be further resolved to quasi-classical *V*_qc_ and interference *V*_I_ contributions, so that:(16)$$V_{\mathrm{inter}}\left({H_{2}}^{+}\right)=V_{\mathrm{qc}}+V_{I}\;,$$
where:(17)$$V_{\mathrm{qc}}=\frac{1}{2}\left[\langle \phi_{a}\left(\zeta \right)|{\hat{V}}_{b}|\phi_{a}\left(\zeta \right)\rangle +\langle \phi_{b}\left(\zeta \right)|{\hat{V}}_{a}|\phi_{b}\left(\zeta \right)\rangle \right]+V_{N}\;$$

The corresponding resolution for H_2_ is somewhat more complex as in addition to the quasi-classical term *V*_qc_ it includes two distinct types of interference terms *V*_I_ and *V*_II_ as well as a sharing contribution *V*_sc_, which accounts for the increase in electron-electron repulsion energy that is induced by electron sharing [58,59].

The computed intra- and interatomic energies of H_2_^+^ and H_2_ obtained using molecular wave functions with optimized AOs, as summarized in Equations (10)–(15) are shown in Figure 6. The interatomic energies are qualitatively the same as those obtained with *ζ* = 1 in Figure 1, in that for both molecules at all distances *R* the interatomic potential energies are repulsive. The interatomic total energy is attractive, except at small distances (less than ~ 0.8 *a*_0_ in H_2_^+^ and ~ 0.6 *a*_0_ in H_2_), due to the interatomic kinetic energies being negative, i.e., attractive, at all separations. The repulsive *total* kinetic and attractive potential energies that are observed as the AOs contract (see Figure 1) are entirely due to the intra-atomic components, as described in Equations (11) and (12) and shown in Figure 6.

### 2.4. The Effects of Interference on Charge Movement and Energies

The interference density, *ρ_I_*, being the difference between the molecular electron density and the sum of the constituent atom densities, reflects the charge rearrangement that occurs on bond formation (see Equations (6) and (7) and Figure 2, Figure 4 and Figure 5). As illustrated in the contour map in Figure 5, *ρ_I_* consists of a region of positive values (*ρ*_+_) in the interatomic i.e., bond region and a region of negative values, the two separated by sheets of zero density. Integration of the charge density over the *ρ*_+_ region results in *Q*_I_, the total electronic charge transferred into the bond. (Note that positive *Q*_I_ corresponds to charge build-up, i.e., the negative sign of the electronic charge is not taken into account). Integration of the analogous kinetic and electron/nuclei energy distributions yields *T*_I__+_ and *V*_I__+_ as well as *V*_I__+_ and *V*_I__−_, the sums of which result in the total kinetic and potential interference energies *T*_I_ and *V*_I_. The former is equal to the interatomic kinetic energy *T*_inter_, while *V*_inter_ includes a quasi-classical contribution in addition to the interference term, as given in Equation (16). The results of the integrations are summarized in Figure 7 that shows the internuclear distance dependence of the various terms discussed above.

As the nuclei move closer together there is a gradual increase in the charge transfer term, reaching a maximum in the equilibrium region, after which there is a sharper decline to zero as the united atom limit is approached. Clearly, the degree of interference, manifested in the amount of charge transferred to the bond correlates with the strength of the bond, i.e., the interatomic component of the total energy of the molecule. Regarding the kinetic/potential contributions to the latter, as expected, the largest bonding contribution is due to the interference component of the kinetic energy, *T*_I_. In both regions, i.e., *ρ_I_*_+_ and *ρ_I_*_−_, the kinetic energy contributions, (*T*_I+_ and *T*_I__−_) to *T*_I_ are negative, because of the interference part of the wave function becoming smoother due to the delocalization of the molecular wave function. In the case of the potential energy the loss of density from around the atoms results in an increase of *V*_I__−_, while the charge transfer to the bond decreases *V*_I+_, although overall the total, *V*_I_, remains positive, i.e., repulsive, at all distances. In summary, bond formation, i.e., the interatomic energy being negative is due to the negative interference kinetic energy that can be traced to the delocalization of the molecular wave function that is very much a quantum effect.

Although the above analysis has been illustrated for H_2_^+^, the same arguments and conclusions apply to H_2_ as well as the covalent bonds of larger diatomic molecules, in particular to B_2_, C_2_, N_2_, O_2_, and F_2_ [58,59]. Using full valence space multiconfigurational self-consistent field (MCSCF) techniques Ruedenberg and coworkers [58,59,83] demonstrated that the basic synergism between intra- and interatomic energy changes in these larger diatomics are very similar to what had been observed in H_2_^+^ and H_2_.

More recently, Ruedenberg and coworkers [97] have developed a quasi-atomic orbital (QUAO) analysis, where the QUAOs are rigorous counterparts to the bond forming hybrid orbitals and allow straightforward analyses of bonding in a molecule, determining bond orders, kinetic bond orders, hybridizations, and local symmetries. Of particular interest are the kinetic bond orders that provide computationally efficient energy-based quantitative estimates of covalent bonding. The method has been applied to a range of large systems, such as xenon containing molecules [98], the disilyl zirconocene amide cation [99], and cerium oxides [100].

### 2.5. Spatial Analysis of the Density and Energy Changes on Covalent Bonding

The reasons for the decrease in the total molecular energy towards shorter bond lengths that result from orbital optimization, are the critical differences between the electrostatic [24,36,37,38,39] theories and Ruedenberg’s kinetic [49,50,51,52,53,54,55,56,57,58,59] theory of bonding. While the orbital exponent changes, as well as the corresponding decrease in the total energy, are modest, the changes in the kinetic and potential components of the binding energy are quite large. The spatial analysis [81,82,83] that we have developed and used for H_2_^+^ and H_2_ seeks to identify in a very direct way the regions where the major density and energy changes occur.

The essential elements of this approach are the study of density and density difference maps of the kinetic and potential energy integrals (as well as density integrals). Thus the essential spatial features and consequences of the electrostatic and the contrasting kinetic interpretations can be directly compared. The method focuses on the spatial dependence of the density and energy integrals, whereby a given property *P* is expressed in terms of contributions as summarized by the following equations in the case of one-electron properties:(18)$$\langle P\rangle =\int \Psi^{*}\hat{P}\;\Psi dxdydz=\sum_{n}P_{n}$$
where:(19)$$P_{n}=\int_{z_{n}}^{z_{n+1}}\left\{\int_{-\infty }^{\infty }\int_{-\infty }^{\infty }\Psi^{*}\hat{P}\;\Psi dxdy\right\}\;dz\;.$$

Full details of the method as well as results obtained for H_2_^+^ and H_2_ re available in our previous papers [81,82].

The effects of orbital optimization are clearly evident in the difference maps, whereby the change in property *P* is defined as:(20)$$\Delta P=P\left(\zeta_{\mathrm{opt}}\right)-P\left(\zeta =1\right).$$

A comparison of the effects of the molecular and atomic contractions, i.e.,
(21)$$\Delta \Delta P=\Delta P_{\mathrm{molecule}}-\Delta P_{\mathrm{atoms}},$$
provides a relative measure of the atomic and molecular contributions of the density and energy changes that occur on covalent bonding.

The molecular contraction results, as defined by Equation (20), computed for H_2_ at the VB level at the internuclear distance of 1.4 *a*_0_, where *ζ*_opt_ = 1.1695, are displayed in Panel I of Figure 8. The results clearly indicate that the effects of orbital contraction on the density of the molecule and the corresponding kinetic, potential and total energy changes are greatest near the nuclei. Numerical estimates of the intra- and interatomic changes are given in our previous work [82]. Approximately, 75 and 78% of the kinetic and potential energy changes, respectively that occur on contraction are estimated to be atomic in nature. While orbital contraction does result in an increase in the density in the interatomic region, the corresponding contribution to the total energy is essentially zero.

Comparison of the effects of contraction in the molecule with those in the free atoms (Figure 8, Panel II) shows that there is a substantial increase in the interatomic region and a corresponding decrease in the kinetic energy of the molecule. The drop in potential energy is however more than cancelled by the increase in the intra-atomic region, resulting in a net antibonding contribution. Noting that if we define the interatomic components of *P* with an arbitrary *ζ* as:(22)$$P_{\mathrm{inter}}\left(\zeta \right)=P_{\mathrm{molecule}}\left(\zeta \right)-P_{\mathrm{atoms}}\left(\zeta \right)\;,$$
then:(23)$$\Delta \Delta P=P_{\mathrm{inter}}\left(\zeta_{\mathrm{opt}}\right)-P_{\mathrm{inter}}\left(\zeta =1\right)\;,$$
i.e., the plots in Panel II of Figure 8 can be interpreted as the effects of contraction on the properties of interest: density, kinetic, potential, and total energies. Thus, the decrease in total energy, brought about by orbital contraction, is due to the drop in the interference component of the kinetic energy. These observations are in complete accord with the findings of Schmidt et al. [58,59], whereby orbital contraction enhances delocalization, i.e., the transfer of electron density to the interatomic region that results in a lowering of the kinetic energy and hence a lower total energy.

The above results clearly demonstrate that orbital contraction affects the density and energy contribution from the immediate neighborhood of the nuclei and are contrary to the notion that the increased stabilization due to orbital contraction is brought about by the interaction between the increased charge density in the bonding region and the nuclei. The model, first suggested by Ruedenberg and co-workers [49,50,51,52,53,54,55,56,57,58,59], that we should consider electron sharing between atoms with contracted densities, i.e., quasi atoms, as the source of covalent bonding, is strongly supported by these studies.

### 2.6. Covalent Bonding without the Virial Theorem: Non-Coulombic Analogues of H_2_^+^ and H_2_

The central plank in the electrostatic theory of bonding is the Virial Theorem [34,35,36,105,106,107] and, therefore, in simple language, bonding occurs because the attractive potential energy stabilizes the molecule in spite of the repulsive kinetic component. Counter-arguments, such as that in 2.1 above, pointing out the effects of scaling, while accepted by some, have not been seen or taken seriously by the majority of the chemical community. An alternative approach is to demonstrate that the existence of covalent bonds is independent of the exact nature of the inter-particle potential and therefore not dependent on the Virial Theorem [34,35,36,105,106,107] as we shall demonstrate below.

A simple alternative to a Coulombic potential is a Gaussian one where:(24)$$V\left(r\right)=q_{1}q_{2}A\exp\left(-\alpha r^{2}\right)\;.$$

Such a potential (where q1 and q2 are the interacting charges) is quite different from a 1/*r* Coulomb potential, inasmuch as the latter has a singularity at the origin and decays quite slowly with *r* as *r* → ∞, whereas a Gaussian potential is short-ranged and harmonic at the origin. The infinite range of the Coulomb potential enables it to support an infinite number of bound states, whereas the Gaussian potentials used in this work support only one bound state. However, from the point of view of bonding what is important is that the potential does give rise to a stable atom and hence molecules. A detailed description of our comprehensive study on the H_2_^+^ and H_2_ systems using Gaussian potentials has been published elsewhere [83], so in this paper only the salient features of the work will be described.

In our previous work [83], we experimented with four different Gaussian potentials, ranging from very weak (α = 0.25, *A* = 0.5) to very strong (α = 2.0, *A* = 8.0), giving rise to atoms with electrons that are bound accordingly, the ground state energies ranging from −0.019 to −1.568 *E*_h_. In this work, we discuss just one of them, namely the (α = 0.5, *A* = 1.5) one, that gives an energy of −0.185 *E*_h_ (in a basis of eight primitive Gaussians with exponents chosen as an even-tempered sequence [108] (0.01(3*^p^*^−1^), *p* = 1,2…8)). The exact energy (with Coulomb potential) is −0.5 *E*_h_. Scaling the resulting AO by *ζ* (by multiplying the Gaussians’ exponents with *ζ*^2^) gives an AO with orbital exponent *ζ,* so minimal basis computations on H_2_^+^ and H_2_ can be carried out, as discussed above, including orbital optimization.

A one-dimensional illustration of the similarities and differences between the Gaussian and Coulomb potentials and the resulting wave functions in H_2_^+^ is provided in Figure 9. The important double well nature of the Gaussian potential is obvious, as are the qualitative differences between the two potentials, in particular the finite depth of *V*_G_ at the nuclei. Consequently, the most striking difference between the wave functions is the absence of the nuclear cusps in Ψ_G_. While the two wave functions are qualitatively quite similar, the question is whether the energetic trends that occur on covalent bonding are essentially the same.

The internuclear distance dependence of the computed energies for H_2_^+^ with the above Gaussian potential is shown in Figure 10. The qualitative trends in the kinetic, potential, and total energies are essentially the same as for the Coulomb potential (Figure 9), although there is a minor but noticeable difference in the behavior of the potential energy. In the case of the Gaussian potential for fixed orbital exponents, i.e., *ζ* = 1, the potential energy initially rises as the internuclear distance decreases, but then it dips to a minimum (that is actually below the atomic value) before becoming strongly repulsive. This behavior is much more accentuated for optimized orbital exponents. In the case of the Coulomb potential, there is no minimum in the potential energy for *ζ* = 1 (see Figure 1). Thus, irrespective of the potential used, molecular binding is almost entirely kinetic in origin when *ζ* = 1. Further, the orbital contraction (*ζ* > 1), that occurs as the internuclear distance decreases, affects the kinetic and potential energies the same way, irrespective of the potential used, resulting in large shifts in both kinetic and potential energies. The corresponding net decrease in the total energy, however, is quite modest. The equilibrium bond lengths are 2.58 *a*_0_ for *ζ* = 1 and 2.47 *a*_0_ for *ζ* optimized to 1.136. The corresponding binding energies are −0.110 *E*_h_ and −0.120 *E*_h_, respectively. Similar trends were observed for the other Gaussian potentials, for both H_2_^+^ and H_2_.

These results are obvious consequences of the strong dependence of the kinetic and potential energy in an H atom as well as both molecules (H_2_^+^ and H_2_) on small variations of the wave function, such as contraction or expansion, irrespective of the nature of the inter-particle potential.

In our previous work [83], where a number of different Gaussian potentials were investigated, we reached the general conclusion that, notwithstanding the marked differences in the distance dependence of the energies of the various potentials, at a fundamental level they all exhibit the same characteristics. Thus, in all cases, for both H_2_^+^ and H_2_, just as for Coulomb potentials, the interatomic component of the total energy, *E*_inter_ is responsible for bonding. The dominant contribution is the interatomic kinetic energy, *T*_inter_, which is equivalent to the interference component of the kinetic energy, *T*_I_. As the latter is a wave mechanical quantity, it follows that covalent bonding is fundamentally a quantum phenomenon, irrespective of the nature of the potential. The intra-atomic total energy, *E*_intra_, is invariably repulsive since the repulsive intra-atomic kinetic energy, *T*_intra_, more than cancels the attraction due to the intra-atomic potential energy, *V*_intra_. The interatomic potential energies, *V*_inter_, are, however, mostly repulsive.

Hence, we can safely conclude that the mechanisms of covalent bonding in H_2_^+^ and H_2_ are the same, irrespective of the nature, Coulombic or Gaussian, of the inter-particle potential. Thus, the Virial Theorem [34,35,36,105,106,107] is not required for bond formation and, in the case of the Coulombic potentials, it is not the cause of bond formation.

### 2.7. The Dynamics of Electron Delocalization in H_2_^+^: Time-Dependent Description

All too often students visualize and think about covalent bonding as a static phenomenon, where the shared electrons are located in-between the bonded atoms and act as “electronic glue.” This is too literal an interpretation of Lewis structures, where the only role of quantum theory is to replace Lewis’ shared electrons with electron clouds which are localized in the “binding” regions of the molecule. The reality, although it may not be obvious from the standard time-independent calculations, is that electrons are constantly on the move.

In this section, we discuss electron delocalization using the tools and language of time-dependent quantum theory, which is at the heart of the dynamic description of covalent bonding to be discussed more generally in Section 3 below. Our treatment concentrates on H_2_^+^ so as to make the concept of a shared electron in H_2_^+^ particularly transparent.

To keep the analysis as simple as possible, following Feynman [88], we treat H_2_^+^ as a two-state system spanned by a minimal set of normalized AOs, *φ_a_*, and *φ_b_*. The eigenfunctions of the Hamiltonian are the bonding and antibonding MOs, *ψ_g_* and *ψ_u_*, which are:(25)$$\psi_{g,u}={\left[2\left(1\pm S_{ab}\right)\right]}^{-1/2}\left[\phi_{a}\pm \phi_{b}\right]\;.$$

We assume that the nuclei ***a*** and ***b*** are sufficiently far from each other so that the overlap of the AOs can be neglected, i.e., *S_ab_* = 0. The time evolution of any arbitrary time-dependent state |φ(r,t)〉 of this system can be written in terms of the eigenfunctions *ψ_g_* and *ψ_u_* as:(26)$$|\varphi \left(r,t\right)\rangle =\langle \psi_{g}\left(r\right)|\varphi \left(r,0\right)\rangle \exp\left(-iE_{g}t/\hbar \right)|\psi_{g}\left(r\right)\rangle$$
(27)$$\;+\langle \psi_{u}\left(r\right)|\varphi \left(r,0\right)\rangle \exp\left(-iE_{u}t/\hbar \right)|\psi_{u}\left(r\right)\rangle \;,$$
where |φ(r,0)〉 is the initial (localized) state of interest, i.e., at *t* = 0, which we assume to be the AO $|\phi_{a}\rangle$. Thus, Equation (26) can be rewritten in the simple form:(28)$$\;|\varphi \left(r,t\right)\rangle \;=2^{-1/2}\left[\exp\left(-iE_{g}t/\hbar \right)|\psi_{g}\left(r\right)\rangle +\;\exp\left(-iE_{u}t/\hbar \right)|\psi_{u}\left(r\right)\rangle \right].$$

The decay and subsequent variation of the integrated probability density associated with nucleus ***a***, i.e., electron number *n_a_*, in the Hilbert space spanned by $\left\{|\phi_{i}\rangle \right\}$ is described by the projection:(29)$$n_{a}\left(t\right)=\langle \phi_{a}|\varphi \left(r_{a},t\right)\rangle \langle \varphi \left(r_{a},t\right)|\phi_{a}\rangle \;\;\;=\frac{1}{4}\left\{{\left|\exp\left(-iE_{g}t/\hbar \right)\right|}^{2}+\;{\left|\exp\left(-iE_{u}t/\hbar \right)\right|}^{2}\;\right\}\;\;+\frac{1}{4}\left\{\exp\left[i\left(E_{u}-E_{g}\right)t/\hbar \;\right]+\exp\left[-i\left(E_{u}-E_{g}\right)t/\hbar \right]\right\}\;\;\;=\frac{1}{2}+\;\frac{1}{4}\exp\left(i\Delta Et/\hbar \right)+\frac{1}{4}\exp\left(-i\Delta Et/\hbar \right)$$
(30)$$n_{a}\left(t\right)=\frac{1}{2}\left[1\;+\cos\left(\Delta Et/\hbar \right)\right]\;,$$
where:(31)$$\Delta E=E_{2}-E_{1}$$

Thus, *n_a_* is a periodic function of time with a periodicity of 2πℏ/ΔE. The corresponding transfer (tunneling) rate $\tau^{-1}$ (of the electron from one nucleus to the other) is therefore predicted to be:(32)$$\tau^{-1}=\frac{\Delta E}{\pi \hbar }\;$$

This is the standard formula for transfer rate in simple two level systems. Further:(33)$$\Delta E\left(R\right)=E_{u}\left(R\right)-E_{g}\left(R\right)=2\left[E_{u}\left(R\right)-E_{H}\right]$$
(34)$$\;=2\left[E_{H}-E_{g}\left(R\right)\right]=2B\left(R\right)$$
where B(R) is the (positive) binding energy of H_2_^+^. We have found that Equation (25), and hence Equations (33) and (34), are applicable at $R\geq 4.5a_{0}$. At such distances therefore the electron transfer rate has a direct dependence on the binding energy:(35)$$\tau^{-1}=\frac{2B\left(R\right)}{\pi \hbar }\;$$

At shorter distances, where ΔE(R)>B(R), the transfer rate rapidly becomes larger than that given by Equation (35).

By way of illustration the computed transfer rates at a number of internuclear separations are shown in Figure 11. At *R* = 8 *a*_0_, the height of the Coulomb barrier is −0.5 *E*_h_, exactly the same as the energy of an H atom. Electron transfer at larger distances therefore occurs by tunneling. As the distance becomes smaller the transfer rate rapidly increases, as expected.

In Figure 12, we showed the probability densities calculated from the electron numbers *n_a_* and *n_b_* of Equation (31) at *R* = 8*a*_0_, for a number of different times. Starting with the electron fully localized on nucleus ***a***, i.e., with density ${\left|\phi_{a}\left(r\right)\right|}^{2}\;$, at subsequent times (200, 350, 450, 550, and 700 au) more and more of the density appears on nucleus ***b***, until at *t* = 900 au the transfer of density to ***b*** is complete.

The important point, which the above analysis illustrates, is that the shared electron that corresponds to the covalent bond in H_2_^+^ is not localized. Assuming that at an initial time *t* = 0 the electron is associated with nucleus ***a***, we see that it does not remain localized on that nucleus, but moves to nucleus ***b*** and back, i.e., executes an oscillatory behavior between the atomic centers. This means that the ground state, which we normally refer to as stationary, is not localizable except in the limit of infinite separation *R*, where there are two states of equal energy which we can represent as left or right localizable, if we prefer. For any finite *R* the ground and first excited states are split by an energy representing the rate of interatomic electron transfer of a localized electron. This means that delocalization (electron sharing), as used in the Hellmann–Ruedenberg [40,49,50,51,52,53,54,55,56,57,58,59,83] energy analysis of covalent bonding, is a *dynamical* process. The shared electron, in addition to moving in the proximity of one nucleus, will transfer to the other nucleus and back with a well-defined periodicity. An average over all possible phases of that motion leaves the electron probability density time-independent (stationary) but the electron is incessantly moving, as we know from its kinetic energy. Further, the rate of electron transfer in the large distance regime, i.e., approximately for $R\geq 4.5a_{0}$, is determined by the antibonding–bonding energy splitting, which is predominantly kinetic in character, since $T_{u}-T_{g}>E_{u}-E_{g}$. Indeed, the effect of potential energy is to reduce the electron transfer rate, since at any distance $V_{u}-V_{g}<0$. These observations underpin our previous analyses, where we deduced that the kinetic energy has a critical role in the phenomenon of covalent bonding.

The spatial component of the Heitler–London wave function (Equation (2)) is symmetric with respect to the permutation, i.e., interchange, of the two electrons. Thus each electron has equal probability of being described by AO *φ_a_* or *φ_b_*, i.e., “being associated with” or “being on” either nucleus ***a*** or ***b***. The total (one-electron) density *ρ*(**r**) is actually readily shown to be:(36)$$\rho \left(r\right)=\;{\left(1+S_{ab}^{2}\right)}^{-1}\;\left[\phi_{a}^{2}+\phi_{b}^{2}+2\phi_{a}\phi_{b}S_{ab}\right]\;.$$

Thus, the wave function (Equation (2)) and the corresponding density are delocalized and symmetric (with respect to the exchange of nuclei). Moreover, the motion of the two electrons is left-right correlated. That means that, loosely speaking, as one electron moves from ***a*** to ***b***, the other will move from ***b*** to ***a***, i.e., the flip-flop motion of the electrons, as described by Feynman [88], is correlated. If we used a configuration interaction (CI) wave function, that, in addition to the Heitler–London covalent terms (Equation (2)) would include ionic contributions as well, i.e.,
(37)$$\Psi_{\mathrm{CI}}=c_{\mathrm{cov}}\Psi_{\mathrm{cov}}+c_{\mathrm{ion}}\Psi_{\mathrm{ion}}\;,$$
where *c*_cov_ and *c*_ion_ are variational constants and:(38)$$\Psi_{\mathrm{ion}}=\left[2\left(1+S_{ab}^{2}\right)\right]{\;}^{-1/2}\left[\phi_{a}\left(1\right)\phi_{a}\left(2\right)+\phi_{b}\left(1\right)\phi_{b}\left(2\right)\right],$$
the time-dependent wave function would represent a flip-flop motion composed of a mixture of in-phase (ionic) and out-of-phase (covalent) electron exchanges between the atomic centers. In other words, it would allow for the possibility of both electrons being on nucleus ***a*** or ***b*** at the same time.

## 3. The Quantum Dynamical View of Covalent Bonding

Despite the rigorous analysis and arguments of Ruedenberg et al. [49,50,51,52,53,54,55,56,57,58,59,83] during the last 58 years, and its acceptance by many prominent scientists [60,61,62,63,64,65,66,67,68,69,70,71,72,73,74,75,76,77,78,79,80,81,82,83,84,85,86,87], the theory of covalent bonding, as developed by Ruedenberg et al., is not universally known or accepted. Even among experts there is continuing debate about the origin of bonding and Chemistry textbooks often present simplistic outdated views of the physical origin of bonding or avoid controversy by only presenting facts and the simplest quantum chemistry that can reproduce them.

The main reason for the apparent confusion and controversy is, we believe, the quantum mechanical nature of the mechanism. In classical mechanics, the search for a ground state is a search for the geometrical configuration of minimum (potential) energy. In quantum mechanics we search for a ground state that’s described by a wave function, i.e., the state is effectively diffuse over geometries and has both potential and kinetic energy. While the potential energy is minimized by a localized state, the kinetic energy is minimized by maximizing spatial diffuseness. Moreover, the diffuseness included in the ground state must be dynamically connected. All this is accounted for in the search for a lowest energy solution of the Schrödinger equation, e.g., by the finite basis set method in Hartree–Fock SCF or correlated calculations. It is easier, given our mainly classical experiences, to envisage the minimization of potential energy than a total energy with a kinetic component, given its complex relation to dynamics. This may explain some of the difficulty experienced in understanding covalent bonding, which, we claim, is related to reduction of kinetic energy and to interatomic electron dynamics in molecules.

### 3.1. The Quantum Mechanics and Dynamics of Atoms

The role of quantum mechanics and its effect on electron dynamics is deeply embedded in chemistry. For a long period in its early history chemistry was dominated by the search for elements and its greatest early achievement was their ordering into the periodic table. This was incredibly important because the properties of these atoms were found to be strongly variable, yet well predicted by each atom’s place in the table. The periodic table, thus, displays clear and well defined trends in atomic reactivity [75] and related properties, i.e., chemical periodicity [109]. The earliest discussions of atomic reactivity and molecule formation were in terms of oxidation and reduction, i.e., the transfer of electrons among bonded atoms according to valence numbers that were predictable from the place of an atom in the periodic table. The whole structure of the table was related to the particularly stable inert gas atoms and it was realized that molecule formation was contingent on the participating atoms, by electron transfer (once electrons as particles carrying a well-defined charge were identified) or sharing, approaching inert gas electronic structure, albeit around atomic centers with varying “non-inert” nuclear charge.

It is clear, in hindsight, that the periodic table represented to the chemists an empirical form of quantum mechanics long before this subject was born at the hands of Planck, Einstein, Bohr, Schrödinger, Heisenberg, and many others. The structure of the periodic table, as we now know, is due to quantum mechanics and its response to the spherical symmetry of the Coulomb attraction between electrons and nucleus in the atom. This symmetry results in conservation of angular momentum which, together with the approximate validity of the independent electron model (mean field or SCF approximation) for many-electron atoms, produces the shell structure reflected in the periodic table. Thus the conservation of angular momentum and spin are *dynamical constraints* on the motion of electrons in atoms which produce strain, i.e., increase in the energy of an atom, which in turn translates into its reactivity.

The presence and nature of these strains are reflected in the degeneracy of the ground state of a given atom as seen in the Aufbau rules or in the full Russel–Saunders or *j,j* coupling degeneracy [110]. For example, the ^3^*P*, ^4^*S*, and ^3^*P* states of C, N, and O, respectively are 9-, 4-, and 9-fold degenerate (according to the Russel–Saunders method in the absence of spin–orbit coupling). Degeneracy (number of states *d* > 1 of the same energy eigenvalue), or near-degeneracy (small energy splitting between a set of energy eigenstates), are closely associated with dynamical constraints such as conservation laws or related barriers to motion, and thereby with reactivity in quantum mechanics. Reactivity is low (or high) if it takes much (or little) energy to add, remove or restructure electrons from the initial configuration by some external agent. If the ground state electronic structure of an atom is degenerate and/or near-degenerate then the ability to respond to an external agent, e.g., in the form of an approaching other atom, is proportionately enhanced [75,79,80].

We recall that if the electron-electron repulsion is neglected in atoms we get a hydrogenic model of atoms in which the one-electron states are defined by the quantum numbers *n, l, m, and s*, where the state energy ε*_n_* (in *E*_h_), in addition to the atomic number *Z* (i.e., nuclear charge *Z*|e|), only depends on the principal quantum number *n*:(39)$$\varepsilon_{n}=-\frac{Z}{2n^{2}}\;,\;n=1,2,3,\ldots$$

The degeneracy is then 2(2*l* + 1) summed from *l* = 0 to *n* − 1 which yields *d_n_* = 2*n*^2^. Thus, we have shells of 2, 8, 18, … degenerate states. Hydrogenic atoms are hugely degenerate and reactive.

Allowing for the electron-electron repulsion in a simple mean field (SCF) approximation we retain the approximation of independent electrons but the potential they move in is no longer purely Coulombic but screened Coulombic. A simple but quite good approximation for such potential is (in *E*_h_) is:(40)$$V_{\mathrm{atom}}\left(r\right)=-\left[1+\left(\eta_{Z}-1\right)\exp\left(-\kappa r\right)\right]r^{-1}\;=-Z_{\mathrm{eff}}r^{-1},$$
where an optimized *κ* (in a_0_^−1^) in the exponential screening term would be ~2 and *η_Z_* would be the appropriate nuclear charge *Z*. Looking at this potential as a Coulomb potential with a separation dependent nuclear charge *Z*_eff_(*r*), we note that this effective charge would be *Z* for small *r* but would decrease to 1 (i.e., account for complete screening) for large *r*. Now the energy of the one-electron states would depend on the angular momentum quantum number *l* but not on *m* and *s*. We would get, as empirically known and used in the “Aufbau rules”, the energy ordering and degeneracies, i.e., maximum occupancy (in parentheses) by electrons of both spin directions, i.e., $m_{s}=\pm \frac{1}{2}\;.$
(41)$$\varepsilon_{1s}\left(2\right)<\varepsilon_{2s}\left(2\right)<\varepsilon_{2p}\left(6\right)<\varepsilon_{3s}\left(2\right)<\varepsilon_{3p}\left(6\right)<\varepsilon_{4s}\left(2\right)<\varepsilon_{3d}\left(10\right)\;\ldots$$

The degeneracy of 2(*2l +* 1) is now restricted to the *n*, *l*-subshells and there is growing dependence of the energy on *l*, the amount of rotation in the motion. We note that rotation tends to keep the electron away from the stronger attraction at smaller *r*. This degeneracy of subshell energies results in significant degeneracy of the corresponding electronic structures of the atoms which in turn reflects on the reactivity of the atoms [79], still far less than that of hydrogenic atoms. Adding Fermi correlation (often called exchange correlation—see below) between electrons of the same spin splits the Aufbau states (according to Hund’s rule) favoring high spin states. This splitting is generally (but not always) of smaller energy than that between subshells with less effect on reactivity but all lifting of degeneracy and dispersal of nearly degenerate states serves to stabilize the system with respect to reactivity. Only the inert gas atoms He, Ne, Ar, … have nondegenerate and chemically very stable ground state electronic structures.

Without constraints in the form of spin and angular momentum conservation, dynamics would couple the degenerate states and increase the diffuseness, and lower the kinetic energy, resulting in nondegenerate ground states for all atoms. Near-degeneracies have a similar but lesser effect in proportion to the energy splitting. Thus the stabilities of the ^1^*S* atoms Be and Mg, atoms with nondegenerate ground states, are not comparable with those of the inert gases because the energy separation between the ground and first excited state is small for the former (within the same shell) but large for the latter (excited state in a higher shell).

### 3.2. The Quantum Dynamics of Molecule Formation

As discussed above, atomic instability (alternatively called strain or reactivity) is due to the presence of dynamical constraints which localize the electron dynamics and correspondingly the energy eigenfunctions which reflect this dynamic localization [79,80]. Looking now at the situation in the formation of H_2_^+^ we see a closely related bonding mechanism. At large separations *R,* the electron is in one or the other of the two potential wells around the protons. Even if we conserve the spin there is a two-fold energy degeneracy due to the lack of motion between the protons. By decreasing the separation, a kinetic coupling sets in which, as we have seen in Section 2 above, gradually lifts the degeneracy and stabilizes the molecular ion. A dynamical constraint in the form of a potential barrier to interatomic motion is lifted as *R* decreases. The result is delocalization and stabilization proportional to the frequency of interatomic electron oscillation. This has been shown in detail in the preceding section for the standard minimal atomic basis set of two non-orthogonal functions. Thus bonding in H_2_^+^ is due to the lifting of a left or right localization of the electron and a corresponding lifting of degeneracy which turns into near-degeneracy of decreasing significance (larger energy splitting) as *R* decreases. We now consider what happens to this mechanism as we go to two electrons in H_2_ and then, in a simplified manner, to larger systems with three or more atoms.

#### 3.2.1. The Correlation Mechanism

In our treatment of H_2_ above, we have encountered, but not elaborated on, the electron correlation mechanism which has caused some complication in the treatment of molecular electronic structure and bonding over the years. By choosing the VB wave function for H_2_ we have accounted for the correlation mechanism in perhaps the simplest way fully consistent with its nature. Our purpose here is to explain why the simpler independent electron (SCF) approximation is not adequate in the discussion of covalent bonding and how the correlation mechanism alters the dynamical mechanism of covalent bonding.

The general statistical meaning of “correlation” is that two or more variables are not independent but the probabilities of one of the variables are dependent on the state of the other. The full multi-variable probability density then cannot be written as the product of probabilities of each variable. In quantum chemistry, one talks of independent electrons or correlated electrons but the former are distinguished by the ability to describe them by a wave function in the form of a (Slater) determinant of linearly independent (usually orthonormal) spin orbitals. Such a determinant will satisfy the Pauli principle of fermion statistics. This is the foundation of the Aufbau method of constructing atomic ground states from atomic spin orbitals. It should be noted that independent electrons in this Aufbau scheme can still show correlation in the statistical sense due to the fermion statistics and the determinantal wavefunction, but, as discussed above and elaborated below, this is referred to as Fermi correlation or exchange correlation and often not included in the term “correlation.” Fermi correlation operates only between electrons of the same spin and it is therefore not present in the singlet ground state of H_2_. The other type of correlation, unfortunately, is often not further specified, and we shall refer to it as Coulomb correlation. It is due to the Coulombic repulsion between the electrons. It acts in a pairwise fashion between all electrons but its effect is generally strongest between electrons of different spins because electron pairs of the same spin are already kept apart by Fermi correlation so only weaker long range forces act to further distance electrons of the same spin from each other.

In the singlet ground state of H_2_ we have a pair of electrons of opposite spin, so Coulomb correlation plays a major role. The VB wave function we have used in Section 2 is constructed using non-orthogonal atomic basis functions, i.e., as a combination of covalent two-electron Slater determinants, in order to keep the two electrons apart and reduce the Coulomb repulsion between them. (It also ensures correct dissociation into H atoms). This type of Coulomb correlation included in the VB wave function may indeed be called an electrostatic bonding mechanism. It certainly plays an important role, which is why we chose to include it in our standard model for H_2_ above, but it is not the key to bonding as we shall see below.

The VB approach has been very popular as an empirical tool for the understanding of bonding and molecular structure, but in ab initio applications the non-orthogonalities encountered in molecules larger than H_2_ have long dissuaded most computational chemists from using it. Instead molecule formation and structure has been approached computationally by first assuming independent electrons in the sense that a SCF wave function in the form of a determinant of orthogonal spin orbitals is assumed and determined by some (orbital) optimization procedure, at a chosen molecular geometry [28,29]. The computation of correlation effects is then carried out in the basis set of the orthogonal occupied and unoccupied (virtual) SCF molecular spin orbitals [111]. This second step (of accounting for Coulomb correlation) is done in a basis of determinantal configurations formed by inserting virtual orbitals in the ground state SCF determinant in place of originally occupied orbitals. These “excited” determinants (single, double, … excitations) then form (in addition to the SCF reference state) a many-electron basis set in which the Hamiltonian can be diagonalized to find a correlated molecular ground state. The basic term for this method is “configuration interaction” (CI) [111,112] but there are many insightful variations on this basic scheme [111,113].

Following the usual SCF + CI approach to the ground state of H_2_ the first SCF step in the minimal basis amounts to assigning one pair of electrons of opposite spin to the bonding (*σ_g_*) molecular orbital (MO) of H_2_^+^. (In a minimal basis calculation on H_2,_ there is no SCF orbital optimization process). The electrons are moving independently so, in comparison with H_2_^+^, the strength of the delocalization mechanism and, thus, the bond strength, are expected to nearly double in H_2_. There is an additional electron-electron repulsion in H_2_, even in the case of the correlated VB wave function. The data in Figure 13, where the total energies of H_2_ and H_2_^+^ (relative to their respective dissociated values) are compared, bear this out. The stronger bond in H_2_ also results in a shorter bond length. Comparison of the MO, VB, and CI energy curves of H_2_, as shown also in Figure 13, demonstrates that the VB and CI energies are very close and both theories correctly describe dissociation. The MO curve, while reasonably accurate at around equilibrium and shorter bond lengths, does not have the correct limiting behavior as *R* → ∞.

The reason for the non-physical behavior of the MO energies of H_2_, as dissociation occurs, is obvious on reflection. The independent electron motion will include covalent (electrons on different atoms) and ionic (both electrons on the same atom) configurations with equal probability so there will be substantial inter-electron repulsion as R → ∞. The rate of interatomic electron motion, on the other hand, will approach zero in this limit since there is a wide potential barrier to penetrate by tunneling. Thus the basic covalent bonding mechanism is only dominant enough to yield good bonding at short bond lengths but is weakened at large *R* to be overwhelmed by the inter-electron repulsion arising between independently moving electrons.

In view of this “dissociation error” the key delocalization mechanism, which is given its maximal expression in the SCF MO method, must be modified to a correlated form in order to reasonably describe the whole bonding process as H_2_ is formed from two far separated hydrogen atoms. This is not difficult and can be done well by use of either the VB or the optimized CI wave function which both successfully describe the bonding from small to large bond lengths. The Coulombic correlations present in the VB or CI wave functions correspond to a hole creation mechanism whereby one electron avoids close encounters with the other. Since the Coulomb interaction is of long range, the hole created can vary greatly depending both on the physical character of the system and the ability of the basis set used to resolve its details. Coulomb correlation is a localization mechanism and we expect it to increase the kinetic energy of VB or CI in comparison with MO wave functions in order to lower the potential energy and hence the total energy. This is clearly seen in the data in Figure 14. As expected, the kinetic energy (*T*) is lowest when computed by the uncorrelated MO wave function of H_2_ (in comparison with the correlated VB and CI methods). The opposite holds for the potential energies. The net results are the total energies shown in Figure 13. The Heitler–London (VB) wave function yields energies that are superior (lower) to those obtained at the uncorrelated MO level and are only slightly worse than the CI ones. We note that the dissociation error implicit in the MO description of the ground state is contained in the excess potential energy in the *R* → ∞ limit.

It may seem remarkable that the VB ground state, which can be written down so readily without any other optimization than the exclusion of ionic configurations on physical grounds, can work so well. One should note in this connection that the VB method has a subtle basis set dependence. It is dependent on the overlap (non-orthogonality) of the atomic basis set used and is not invariant to a rotation of basis within the same space as originally spanned. This can be seen by constructing a VB-like wave function in terms of Löwdin orthogonalized AOs [114], *ϕ_a_* and *ϕ_b_*, related to the original non-orthogonal AO basis, *φ_a_* and *φ_b_*, via the transformation [80]:(42)$$\varphi_{a}=N\left(\phi_{a}-\mu \phi_{b}\right)\;,\;\varphi_{b}=N\left(\phi_{b}-\mu \phi_{a}\right)\;.$$
The constants are given by:(43)$$\mu =\frac{1-{\left(1-S_{ab}^{2}\right)}^{1/2}}{S_{ab}}\;,\;N={\left(1-2\mu S_{ab}+\mu^{2}\right)}^{-1/2}\;,$$
where *S_ab_* is the overlap integral $\langle \phi_{a}|\phi_{b}\rangle$.

At this point, it is helpful to express the VB and CI spatial wave functions in terms of the bonding and antibonding MOs, *ψ_g_* and *ψ_u_*, respectively:(44)$$\Psi_{\mathrm{VB},\mathrm{CI}}=c_{g}\psi_{g}(1)\psi_{g}(2)+c_{u}\psi_{u}(1)\psi_{u}(2)\;.$$

The coefficients *c_g_* and *c_u_* are determined variationally in a CI calculation, but for a VB wave function they are defined so as to eliminate the ionic contributions. They are:(45)$$c_{g}\left(\mathrm{VB}\right)=\left(1+S_{ab}\right){\left[2\left(1+S_{ab}^{2}\right)\right]}^{-1/2},\;c_{u}\left(\mathrm{VB}\right)=-\left(1-S_{ab}\right){\left[2\left(1+S_{ab}^{2}\right)\right]}^{-1/2}.$$
The VB-like wave function (in terms of the orthonormalized orbitals) is, thus:(46)$$\Psi =2^{-1/2}\left[\varphi_{a}(1)\varphi_{b}(2)+\varphi_{b}(1)\varphi_{a}(2)\right]=2^{-1/2}\;\left[\psi_{g}(1)\psi_{g}(2)-\psi_{u}(1)\psi_{u}(2)\right],$$
which agrees with the true VB wave function in the dissociation limit as *S_ab_* → 0. The potential, kinetic and total energy curves of this proposed ground state wave function are shown in Figure 15. We see that, while the potential energy curve is favorable to bonding, all semblance of covalent bonding by delocalization and kinetic energy lowering has been lost and the total interaction is repulsive. It is not difficult to show that the use of an orthogonalized atomic basis set eliminates the flow of electrons between atomic centers in the molecule. In doing so the key mechanism of covalent bonding is removed and it does not help that the correlation produces a bonding potential energy. This shows clearly: (i) the dominant role of interatomic electron motion and (ii) the dependence of the traditional VB method on the non-orthogonality (overlap) of the atomic basis set which allows correlated electrons to flow between atomic centers.

It is interesting to note here that the failure of a Heitler–London type wave function in terms of orthogonalized atomic orbitals (OAOs) was discussed by Slater [115] in 1951. He noted that the seemingly advantageous OAOs actually eliminated the minimum in the bond energy curve making it purely repulsive. He showed that ionic configurations were needed to recover the bonding, i.e., CI, and noted that, with respect to the energy, the resulting improvement over the original Heitler–London wave function is minimal. Slater did foresee more difficulties overcoming the non-orthogonality problems of the VB approach to molecular electronic structures and properties. What we have added here is an exposure of the close connection of the atomic orbital overlap to the dynamical mechanism of covalent bonding, i.e., the correlated interatomic electron motion which is captured in the original but eliminated in the OAO extension of the Heitler–London wave function.

We emphasize that in the case of minimal basis calculations on H_2_ (as done here) the basis set dependence discussed above is limited to Heitler–London wave functions of Σ_g_ symmetry. Thus, the MO and CI wave functions and energies are invariant to any linear transformation of the MOs, such as orthogonalization. In the case of the MO wave function, we noted that as the *σ_g_* and *σ_u_* MOs themselves are invariant to any linear transformation, the MO wave function $\psi_{g}(1)\psi_{g}(2)$ is similarly invariant. As the CI wave function is determined variationally in the basis of all Slater determinants of Σ*_g_* symmetry (only two in the current situation, as given by Equation (44), it is also invariant under any linear transformation of the one-electron basis.

#### 3.2.2. Exchange Correlation, Pauli Repulsion, and Extended Delocalization

We have seen so far in our study of H_2_^+^ and H_2_ how covalent bonding fundamentally arises by the facilitation of delocalization of electron motion over the two atomic centers. The rate of the interatomic transfer is directly related to, and therefore a measure of, the strength of bonding. At the same time two localization mechanisms, orbital contraction and electron correlation (in H_2_ and multi-electron molecules in general), operate to reduce the potential energy at some lesser cost in increased kinetic energy. These mechanisms persist for larger molecules as well, plus three more that contribute to covalent bonding in larger systems: (a) Fermi (exchange) correlation, (b) Pauli repulsion, and (c) extended delocalization (over more than two atoms).

We first consider the Pauli repulsion and Fermi, or exchange, correlation mechanisms, both of which arise when two or more electrons with the same spin are present. We know from the simple Aufbau picture of atomic and molecular structure that there is a kind of repulsion between electrons of the same spin going beyond the usual electrostatic repulsion and forcing them to occupy different orbitals in the usual independent electron models. The origin of this exclusion of double orbital occupancy for electrons of the same spin direction is the need for the total wave function to be antisymmetric with respect to any pairwise electron interchange which in turn follows from the fact that electrons, having odd spin, are fermions that obey Fermi–Dirac statistics. This is vital to the whole Aufbau picture and to the behavior of electrons in atoms, molecules, and solids. The kind of “statistical repulsion” which follows is often referred to as Pauli repulsion.

It is of great importance in connection with bonding and molecule formation to realize that the obvious cost in energy of the Pauli repulsion also brings a reduction in energy due to the Fermi correlation mechanism which reduces repulsion between electrons by a “hole creation mechanism” due to the antisymmetry of the wave function. It is readily seen that, e.g., the spatial component of the lowest energy triplet wave function of H_2_:(47)$${}^{3}\Psi_{u}=2^{-1/2}\left[\psi_{g}(1)\psi_{u}(2)-\psi_{u}(1)\psi_{g}(2)\right]={\left[2\left(1-S_{ab}^{2}\right)\right]}^{-1/2}\left[\phi_{a}(1)\phi_{b}(2)-\phi_{b}(1)\phi_{a}(2)\right]\;,\;$$
vanishes if electrons 1 and 2 are both at the same point in space. As previously, *ψ_g_* and *ψ_u_*, are the orthonormal bonding and antibonding MOs constructed from the 1s AOs, *φ_a_* and *φ_b_*, as specified in Equation (25). Since the wave function is smoothly varying, this means that there is an extended exclusion of the two electrons coming close. This is what is meant by the “Fermi correlation hole” which arises among electrons of the same spin.

The energy of the triplet state is repulsive, as first found by Heitler and London [11]. Thus, as the internuclear separation decreases the overlap integral *S_ab_* becomes larger, and that is largely responsible for the increase in the kinetic energy that is the main cause of the increasingly repulsive nature of the energy. By way of illustration, the energies that were computed in the minimal basis of hydrogenic 1*s* AOs with a fixed exponent of 1, are shown in Figure 16.

The critical contribution of the kinetic energy (*T*) to the repulsive total energy (*E*), shown in Figure 16, is very clear. The small attractive contribution of the *total* potential energy (*V*) at distances larger than ~ 0.8 a_0_ makes a relatively small contribution to the total energy, while at smaller separations the nuclear repulsion term makes an increasingly dominant contribution to the total energy. The distance dependence of the electronic components of the potential energy, i.e., the sum of the nucleus-electron (*V_ne_*) and electron-electron (*V_ee_*) contributions to the potential energy is also shown in the diagram. So as to make the comparison with the kinetic energy more transparent, it is the magnitude of this overall attractive term that is shown, i.e., −(*V_ne_* − *V_ee_*). It should also be noted that the triplet wave function, as given in terms of the AOs *φ_a_* and *φ_b_* in Equation (47), and hence the energy, are invariant to any linear transformation of the occupied orbitals, such as Löwdin or Schmidt orthogonalization (that yield *ϕ_a_* and *ϕ_b_*) or the formation of bonding and antibonding MOs (*ψ_g_* and *ψ_u_*). It is well known that the energy of an electron in an antibonding MO is higher than in the bonding one, due to the higher kinetic energy of the latter. Thus, there is a net increase of antibonding character if both MOs are occupied. Pauli repulsion is often described as the energy penalty associated with the orthogonalization of the occupied orbitals. This association is more subtle (involving a cancellation of attractive electrostatic and repulsive kinetic energy effects) than may be immediately apparent, but, certainly, if the non-orthogonality of the AOs could be neglected then the Pauli repulsion would not appear.

It is our contention that the repulsion seen in triplet H_2_ above can be called “Pauli repulsion”. More generally, the use of the Pauli principle in the construction of atomic and molecular SCF configurations is so familiar that we may forget that it is also responsible for the repulsion between electrons of like spins, due to the requirement of antisymmetry of the total electron wave function. For two interacting atoms this means that the bonding and antibonding MOs that can be constructed from the doubly occupied AOs from different atoms are fully occupied. As during bond formation these nominally non-bonding AOs overlap, i.e., the atomic electron densities are allowed to interpenetrate, the net result is an increase in the electrons’ kinetic energy, which generally more than cancels an electrostatic binding energy and results in an overall repulsion. The SCF interaction energies for the helium, neon and argon dimers, as shown in Figure 17, illustrate the situation. Use of SCF theory is justified for the study of pure Pauli repulsions. Computations at the correlated coupled cluster [CCSD(T)] level of theory have yielded essentially identical results, although they do resolve the small attractive minima [116] due to dispersion.

As in the case of the triplet state of H_2_, Pauli repulsion between electrons of the same spin is an antibonding mechanism which is overwhelmed by a larger covalent bonding mechanism in case of open shell atoms, such as N, O, or F. However, in the case of two rare gas atoms, where the bonding and antibonding MOs are fully occupied, no covalent bonds exist in the sense that the numbers of bonding and antibonding electrons are equal and cancel in the bond order. Pauli repulsion is responsible for the “steric size” of inert gas atoms such as helium, neon, argon etc. It is also responsible in essence for defining the steric size and shape of stable molecules.

For larger covalently bonded molecules than we have studied in detail here, such as N_2_ and F_2_, the Pauli repulsion provides an important antibonding mechanism of kinetic energy character that must be accounted for in any model of the covalent bond. In F_2_, in particular, Pauli repulsion is often claimed to play a very important role in producing a bond strength far weaker than expected given the well-known high reactivity of the fluorine atom. In a fluorine atom, there are six nonbonding valence electrons, as well as two core electrons, all of which will contribute to Pauli repulsion in F_2_, but only one valence electron which can participate in covalent bonding by interatomic transfer. The net result of Pauli repulsion and covalent attraction by electron delocalization is that the bond in F_2_ is unexpectedly weak and the bond length long. In the case of N_2_, there is obviously a stronger bond, due to less Pauli repulsion as well as a total of six bonding electrons resulting in a greater presence of delocalization attraction, yielding the well-known triple bond, one of the strongest bonds known.

The separation of a total bond energy into relevant components among which Pauli repulsion and delocalization stabilization appear, is, however, a delicate matter, given the compensatory nature of kinetic and potential energy effects seen already in our inert gas dimers above. The energy decomposition analysis (EDA) that has been proposed and extensively explored by Frenking and coworkers [87,102,103,104] offers a way of computing individual contributions to the total interaction energy of a molecule relative to its component fragments. It considers bond formation as a result of a three-step process: (1) classical electrostatic interaction, (2) repulsion by application of the Pauli principle, and (3) optimization of orbitals in SCF or DFT that will allow for interatomic delocalization i.e., covalent bonding. While the sum of the individual energies corresponds to the actual bond energy of a molecule, the component energies, including what is regarded as the Pauli repulsion in step 2, are obviously dependent on the decomposition technique used. This form of EDA does not seek to account for all potential–kinetic compensation as we have in our discussion of triplet H_2_ and inert gas dimers above. Nevertheless, the analysis demonstrates, e.g., in the case of N_2_, that the individual electrostatic and kinetic effects are very large but counteracting, leaving the dominant attractive contribution to come from the orbital optimization in the last step that includes full interatomic delocalization. This is consistent with our view that the triple bond of the N_2_ molecule is associated with the delocalization of the six valence electrons of the two nitrogen atoms.

It follows from our consideration of the Pauli principle above that in Hartree–Fock theory, which is the basis for our MO model above, there is an electron correlation mechanism, often called exchange, automatically included with no extra cost in kinetic energy, but it acts only between electrons of the same spin. Thus it has not been encountered in the ground states of H_2_^+^ and H_2_. We can see the effect of this in the triplet state of H_2_ above, where the two electrons are of the same spin and they are kept apart not only by the VB nature of the two configurations, but also by the interference effects between them which keep the electrons apart beyond the Coulomb correlation in the singlet VB ground state. Thus we distinguish between two correlation mechanisms: Coulomb correlation, as in the singlet VB or CI representation of H_2_, and Fermi correlation (or exchange correlation), as in triplet H_2_. The latter is automatically included in any SCF-MO wave function by virtue of the antisymmetric nature of the total wave function, as it is a Slater determinant of spin orbitals. The effect of Coulomb correlation on the total energy is typically smaller (in absolute magnitude) for molecules larger than H_2_, but nevertheless important to the behavior of the valence electrons and therefore to bonding. Coulomb correlation among electrons is the subject of numerous approximations of ever improving accuracy in quantum chemistry [111,112,113].

Finally we shall consider the delocalization mechanism as extended over more than two atomic centers. This is a subject which is well illustrated in the Hückel theory of π-electrons in planar conjugated hydrocarbon molecules [19,20,117]. It is well known experimentally that such molecules, like butadiene, CH_2_=CH-CH=CH_2_, acquire an extra stability by the ability of the π-electrons to delocalize over the whole chain of four carbon atoms rather than only over a pair of carbon atoms as would seem to be the case in our formula for the molecule. This extra stability is well reproduced by the empirical Hückel theory known to most chemists [20,117]. An even better example is the benzene molecule C_6_H_6_ where the extended delocalization of the π-electrons is over a regular hexagon of six carbon atoms [19]. We have recently used the Hückel theory to explain the role of delocalization dynamics in covalent bonding [80]. The salient point is that delocalization extended over more than two atomic centers often, but not always, leads to further stabilization of the molecule. This is exemplified by the aromatic molecules of organic chemistry. The further stabilization by extension of the electron flow from a pair of atoms to a larger molecular network of atoms is due to the same basic preference for diffuseness in a quantum mechanical ground state that is seen in ordinary pair bonding of atoms, such as in H_2_. It is an empirical confirmation of the role played by inter-atomic electron motion, between pairs, in chains or rings, of atoms, in covalent bonding. It is the key to covalent bonding in all its forms.

## 4. Discussion and Conclusions

Quantum mechanics can be elusive. When we generate a solution to the Schrödinger equation to work out a ground state for a molecule and its energy we end up with a wave function for the electrons. On squaring it we obtain a probability density for the distribution of electrons around the atoms. Thus, we can visualize the electron density and see the molecule as a set of joined up fuzzy balls. The ground state is said to be stationary, meaning that the physical properties like probability densities and electron densities are constants, i.e., do not change with time. With such a stationary object in mind, “seeing where the electrons are”, it is easy to understand that many chemists have been looking for the origin of covalent bonding in the electron density difference between molecule and its constituent atoms. This readily leads to the bonding models based on electrostatics, that concentrate on electrostatic interactions and the changes that occur as a molecule forms. The Virial Theorem (whereby there is always a decrease in the total potential energy as a molecule forms, despite the repulsive nature of the kinetic energy) appears to support such models as the way to understand covalent bonding. Many great scientists like Slater [36], Feynman [37], and Coulson [24] can be cited in support of the view that covalent bonding is due to favorable electrostatic interactions in the molecule formation.

The great contributions of Hellmann [40] and then Ruedenberg and co-workers [49,50,51,52,53,54,55,56,57,58,59,73,83] were to focus attention on the role of quantum mechanical kinetic energy in the energetic analysis of covalent bonding. The electrons, as fuzzy clouds around the nuclei of the molecule, have kinetic energy, not only (electrostatic) potential energy. The interatomic delocalization that is implicit in molecule formation, i.e., bonding, invariably leads to a lowering of the electrons’ kinetic energy (that more than outweighs the antibonding effect of the potential energy). Hellmann [40] and Ruedenberg [49,50,51,52,53,54,55,56,57,58,59,73,83] argue that the behavior of the electrons’ kinetic energy is the quantum effect that yields covalent bonding. In tandem with the interatomic changes there are intra-atomic energy shifts as well, due to orbital contractions that become prevalent in the equilibrium region. As the changes in intra-atomic components of the kinetic and potential energies on contraction occur in opposition to the interatomic ones, the total (intra- plus interatomic) energy changes are in accord with the Virial Theorem. Ruedenberg has had a long battle to convince the scientific community to agree with this view of covalent bonding, as being due to the interatomic decrease of kinetic energy in response to the interatomic delocalization of valence electrons. The main problem has been with the interpretation of the Virial Theorem [34,35,36,105,106,107], i.e., to accept that it does not refute the kinetic energy argument provided the intra-atomic orbital contraction effects are recognized. The latter effects represent a minor contribution to bonding but are not the origin of it. We entirely agree with this analysis and have consistently supported it in our own research, as summarized in Section 2 above. In particular, we have collaborated with Klaus Ruedenberg [83] to show that the covalent bonding mechanism is not confined to systems of particles interacting by Coulombic potentials and therefore the Virial Theorem [34,35,36,105,106,107] (that holds for Coulombic particles) is not required for bonding, nor is it the cause of bonding in Coulomb systems.

The fact that the kinetic energy is the key to covalent bonding tells us already that bonding is a dynamical phenomenon. We have stressed in our earlier work and above that an understanding of covalent bonding from a dynamical point of view is complementary to that of an energy analysis and helpful in both assessing simplified physical bonding models and in seeing the mechanism in its deepest and most general form [80,81,82]. It is an important feature of quantum mechanics that it offers a number of dualities of representations. Perhaps the particle–wave duality is best known but there is also a coordinate-momentum duality. As it turns out, quantum mechanics also offers a duality of representations in terms of either energy or dynamics, which is important for the understanding of covalent bonding. These representations are also mathematically much closer than one would understand on the basis of classical mechanics. As we see in Equations (26)–(35) that describe the oscillation of the initially localized electron in H_2_^+^, if we know the energy eigenfunctions and their energies, then the time-dependence is trivially predicted for all times. This is a general fact in quantum mechanics and it gives us a direct link between the energy and dynamical representations of a phenomenon. It follows that if we have a non-degenerate ground state which is delocalized, then the electron dynamics is delocalized and, as we have shown above, this gives rise to a covalent bond in H_2_^+^ with a bond strength proportional to the energy spacing between the bonding and antibonding states. In turn, this bond strength is proportional to the frequency of interatomic oscillation of the localized electron of minimal energy.

We have shown that this basic dynamical mechanism of covalent bonding is accompanied by an orbital contraction mechanism and, in H_2_, also by an electron correlation mechanism, which both are of an electrostatic character in that potential energy is decreased at the expense of an increase in the kinetic energy. They are, however, both generally of smaller magnitude and their presence is a consequence of the main dynamical delocalization mechanism without which covalent bonding would not arise. We claim that this picture of one main kinetic and two supportive electrostatic mechanisms, all of them clearly related to the character of electron motion, remains valid for larger covalently bonded molecules.

Feynman [88] first proposed this dynamical mechanism as being the essence of covalent bonding. We have elaborated on this idea [79,80], starting from an analysis of the failure [43,44,45] of the Thomas–Fermi theory [41,42], the first density functional theory, to describe bonding. The clarity afforded by the dynamical analysis has consequences for both simple physical bonding models as well as the most fundamental bonding theory of Chemistry. The best example of the former is the Lewis model [1] with its extension to the VSEPR model of molecular geometry [118,119]. In these models, electrons are arranged as dots around atoms or in-between pairs of atoms. Students easily get the impression that the valence electrons are static charges interacting with static nuclei screened by core electrons, but we must understand that the electrons are never still but always moving and the key to understanding bonding is that the “shared” electrons move between the bonded atomic centers. In the same way the empirical valence bond (VB) picture of atoms [16] connected by Heitler–London type covalent bonds [11] should be understood to be showing us molecular networks of atoms that exchange electrons pairwise through the bonds but may also allow further delocalization, i.e., valence electron flow over chains or rings of atoms.

Finally, the dynamical mechanism of covalent bonding provides a link to the basic concept of atomic reactivity which is a necessary prerequisite for bonding and molecule formation [75,79]. Atoms are reactive because their ground state structures are strained by dynamical constraints in the form of conservation laws for spin and angular momentum. Such strains are minimized in inert gases. Interatomic electron motion in molecules allows an approach to inert gas configurations, as envisaged in the early Lewis–Kossel–Langmuir theories [1,4,5,6,7,8]. Seeing bonding from the two seemingly different aspects, energy and dynamics, we recognized that both provide accurate and complete representations of the quantum mechanical reality. However, it is the combination of the two that allows the covalent bonding mechanism to be most completely understood and described.

## Figures and Tables

**Figure 1 molecules-25-02667-f001:**
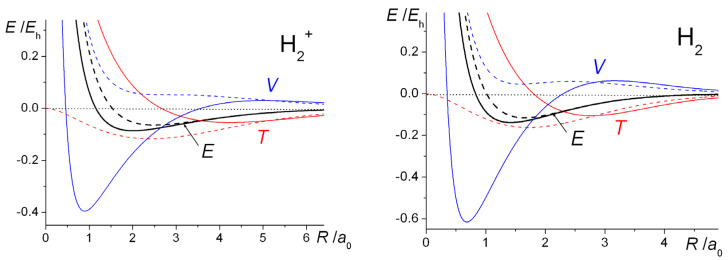
Energies of H_2_^+^ and H_2_ (valence-bond (VB)) (total energy *E*, kinetic energy *T*, potential energy *V*, all relative to the H and H + H atoms respectively), computed using minimal H 1*s* atomic orbital (AO) basis with exponent *ζ* optimized (full lines) and fixed at the H atomic value of 1 (dashed lines).

**Figure 2 molecules-25-02667-f002:**
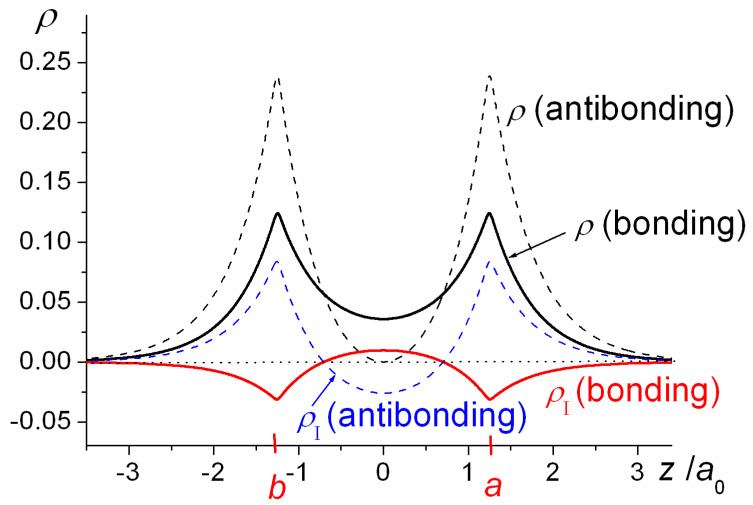
Total electron densities (*ρ*) of bonding (solid black line) and antibonding (dashed black line) states of H_2_^+^ at *R* = 2.5 *a*_0_ with *ζ* = 1 and corresponding interference contributions (*ρ_I_*), shown in red and blue, respectively, as functions of internuclear coordinate *z*.

**Figure 3 molecules-25-02667-f003:**
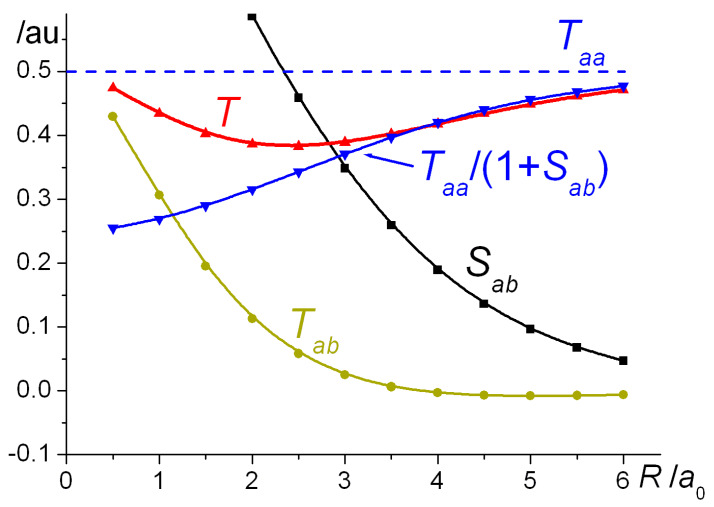
H_2_^+^(*ζ* = 1): Bond length dependence of the overlap *S_ab_*, kinetic energy matrix elements *T_aa_* and *T_ab_* and the contribution *T_aa_*/(1 + *S_ab_*) to the total kinetic energy *T*.

**Figure 4 molecules-25-02667-f004:**
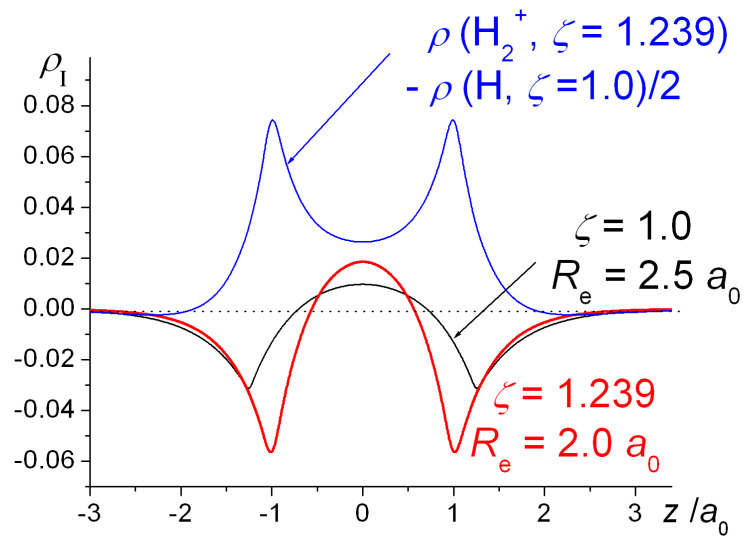
Interference density in H_2_^+^ for optimized exponent *ζ* = 1.239 and bond length *R*_e_ = 2.0 *a*_0_ (red line), with nuclei at *z* = ± 1 *a*_0_, compared with interference densities for *ζ* = 1 and *R*_e_ = 2.5 *a*_0_ (black line) and relative to H atoms with *ζ* = 1 (blue line) as functions of the internuclear coordinate *z*.

**Figure 5 molecules-25-02667-f005:**
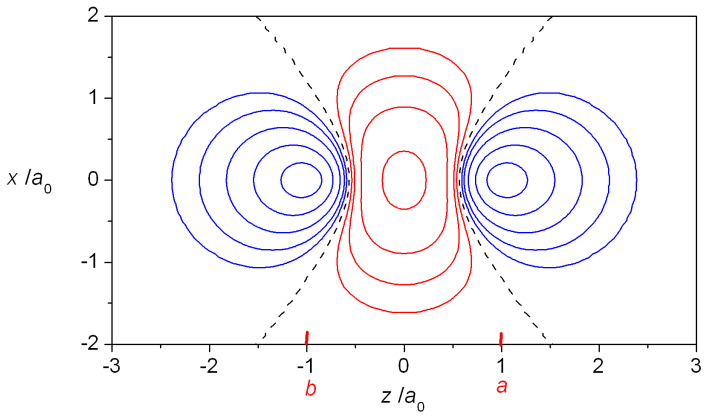
Contour map of the interference density *ρ_I_* in the *xz* plane of H_2_^+^ at *R* = 2.0 a_0_ and *ζ* = 1.239 (where *z* is the internuclear coordinate). The positive/negative contours (*ρ_I_*_+_/*ρ_I_*_−_) in red/blue are ±0.002, ±0.004, ±0.008, ±0.016, and −0.032. The dashed lines represent zero *ρ_I_* contours. The nuclei (*a*, *b*) are at (0, ±1.0 *a*_0_).

**Figure 6 molecules-25-02667-f006:**
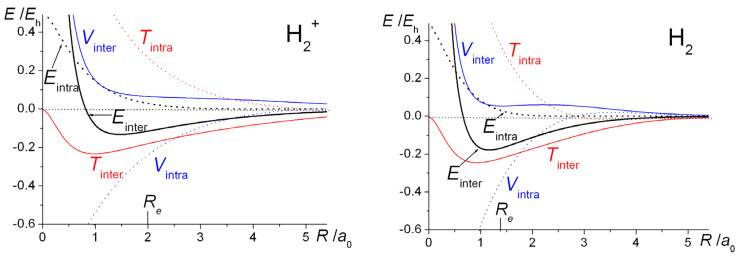
Intra- and interatomic energies of H_2_^+^ and H_2_ from molecular wave functions with optimized *ζ*.

**Figure 7 molecules-25-02667-f007:**
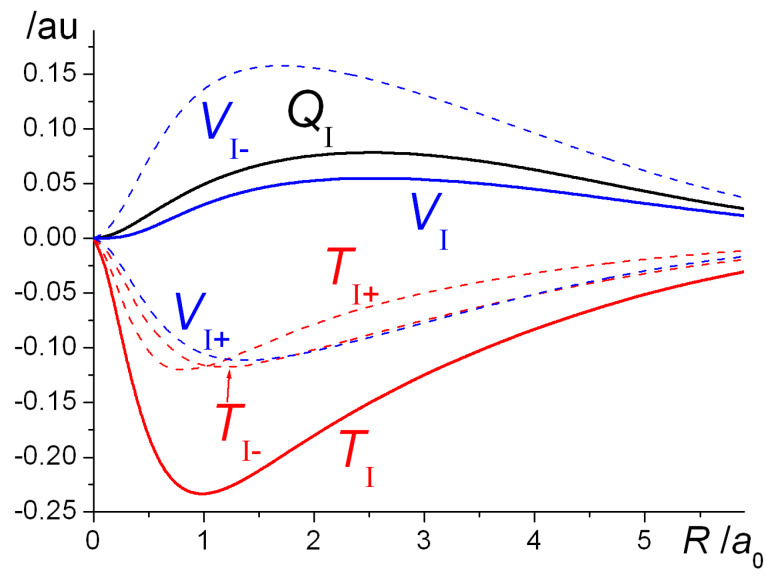
Interference terms for H_2_^+^ with ζ optimized as functions of the internuclear distance *R*. Black line: charge transfer *Q*_I_ into the bond. Red lines: Kinetic interference energies. Blue lines: Potential interferences energies. The contributions *T*_I+_ and *V*_I+_ from the density accumulation region defined by *ρ**_I_*_+_ (See Figure 5). Contributions *T*_I−_ and *V*_I−_ from the density depletion regions defined by *ρ**_I_*_−_ (See Figure 5). The individual contributions are shown by dashed lines, while the totals, shown by full lines, are the sums *T*_I__+_ + *T*_I__−_ and *V*_I__+_ + *V*_I__−_.

**Figure 8 molecules-25-02667-f008:**
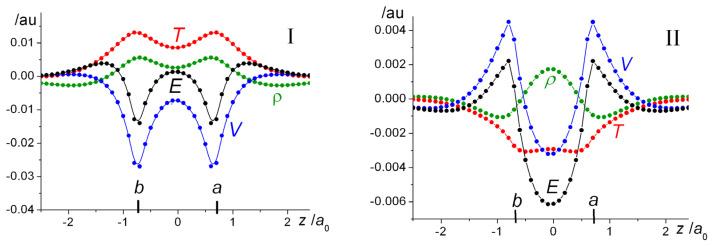
Panel I: Molecular contraction at *R* = 1.4 *a*_0_, i.e., difference maps of integrated densities (ρ) and kinetic (*T*), potential (*V*) and total energies (*E*) between H_2_ with *ζ*_opt_ = 1.1695 and *ζ* = 1.0. Total energy changes (/*E*_h_): Δ*T* = 0.3007, Δ*V* = −0.3342, and Δ*E* = −0.0336. Panel II: Difference between molecular and atomic contractions at *R* = 1.4 *a*_0_ (as defined by Equation (21)). Total energy differences (/*E*_h_): ΔΔ*T* = −0.0670, ΔΔ*V* = 0.0048, and ΔΔ*E* = −0.0623.

**Figure 9 molecules-25-02667-f009:**
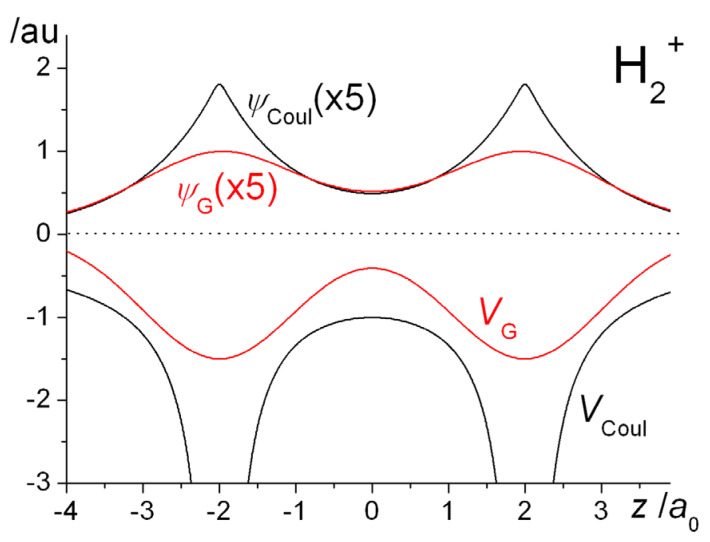
Gaussian (G) (α = 0.5, A = 1.5) and Coulomb (Coul) potentials of H_2_^+^ and the resulting ground state wave functions (with *ζ* = 1 in both cases) at *R* = 4 *a*_0_ as functions of the internuclear coordinate *z* (*x* = *y* = 0).

**Figure 10 molecules-25-02667-f010:**
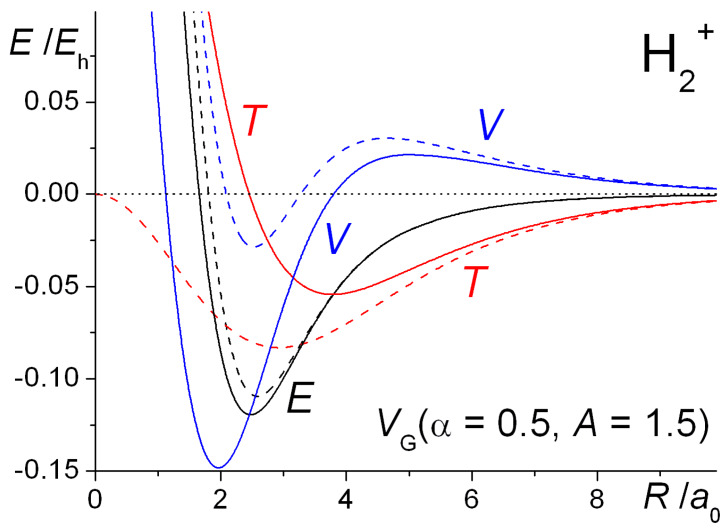
Energies of H_2_^+^ computed with Gaussian potentials (α = 0.5, *A* = 1.5). Results with optimized and fixed (atomic) exponents are shown with full and dashed lines respectively. Total, kinetic and potential energies are shown in black, red, and blue, respectively.

**Figure 11 molecules-25-02667-f011:**
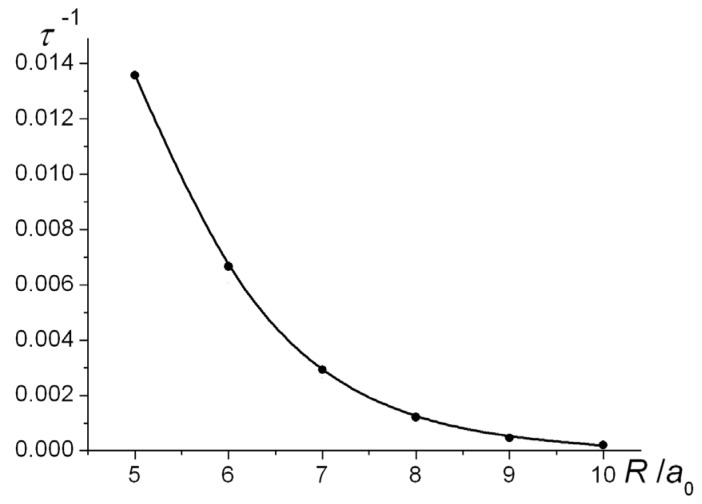
Computed electron transfer rates (from Equation (26)) as a function of the internuclear separation.

**Figure 12 molecules-25-02667-f012:**
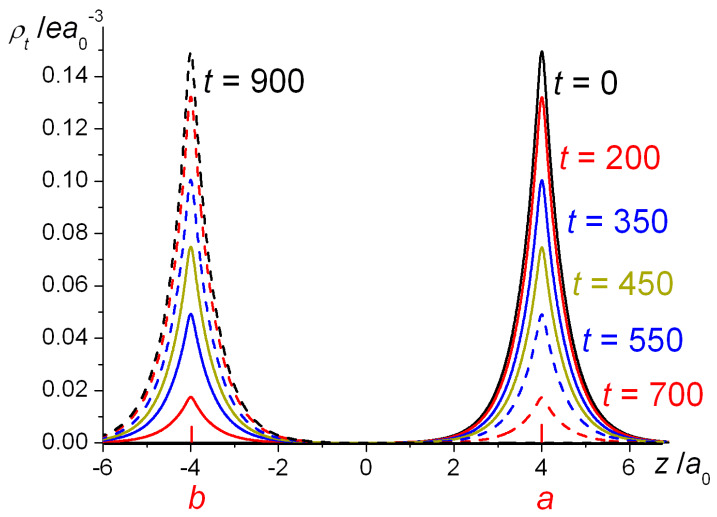
Time dependent H_2_^+^ density $\rho_{t}\left(x,y,z\right)$ with x=y=0 as a function of the internuclear coordinate *z* at selected times (in au).

**Figure 13 molecules-25-02667-f013:**
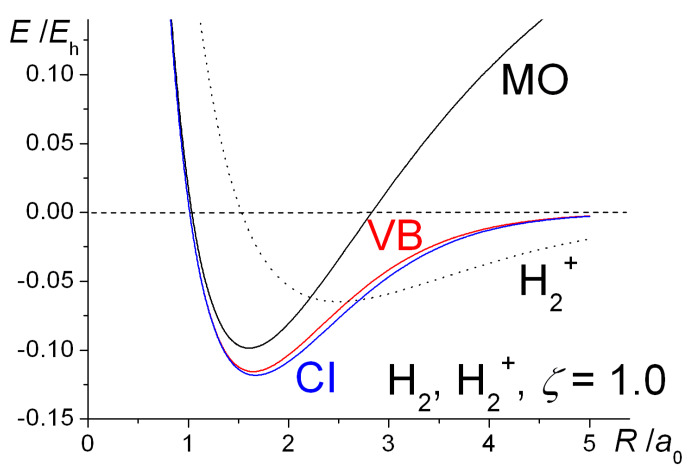
The bond energy curves of H_2_ (relative to the energy of two hydrogen atoms) computed in a minimal basis of two hydrogen AOs with a fixed exponent of 1, as obtained by the MO, VB and CI methods (solid lines), and comparison with the energy curve of H_2_^+^ (dotted line)—all relative to the energy of a H atom.

**Figure 14 molecules-25-02667-f014:**
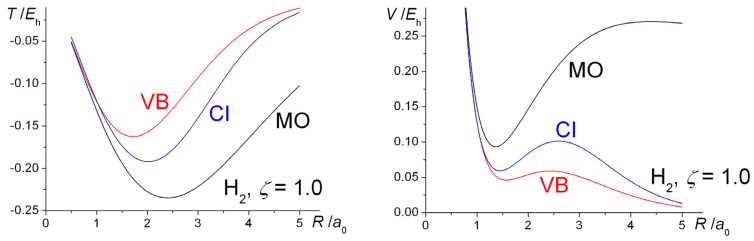
Kinetic (left panel) and potential energies (right panel) of H_2_ computed in a minimal basis of two hydrogen AOs with a fixed exponent of 1, as obtained by the MO, VB, and CI methods (relative to separated limit of two hydrogen atoms). The total H_2_ energies in Figure 13 are the sums of kinetic and potential energies shown here.

**Figure 15 molecules-25-02667-f015:**
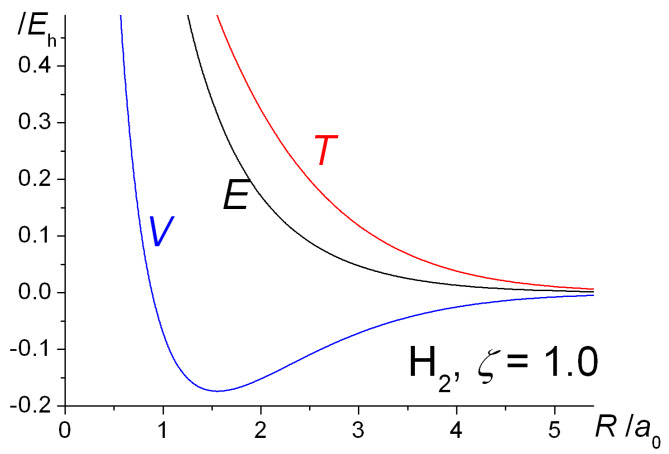
The bond energy of the VB-like ground state formed from Löwdin orthogonalized AOs, as given by Equation (46).

**Figure 16 molecules-25-02667-f016:**
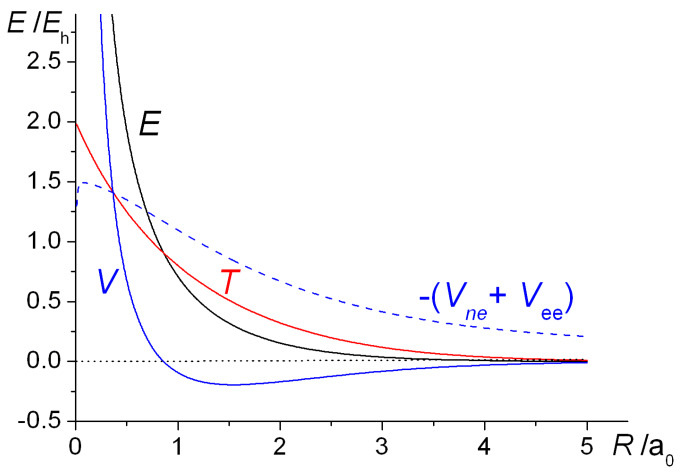
The triplet (^3^Σ_g_) state of H_2_. The total energy (*E*), and its kinetic (*T*) and potential (*V*) components (relative to the energies of two H atoms), are shown in black, red, and blue, respectively. The blue dashed line corresponds to the magnitude of the electronic component of *V*. The energies were computed in the minimal basis of hydrogenic 1s AOs with a fixed exponent of 1.

**Figure 17 molecules-25-02667-f017:**
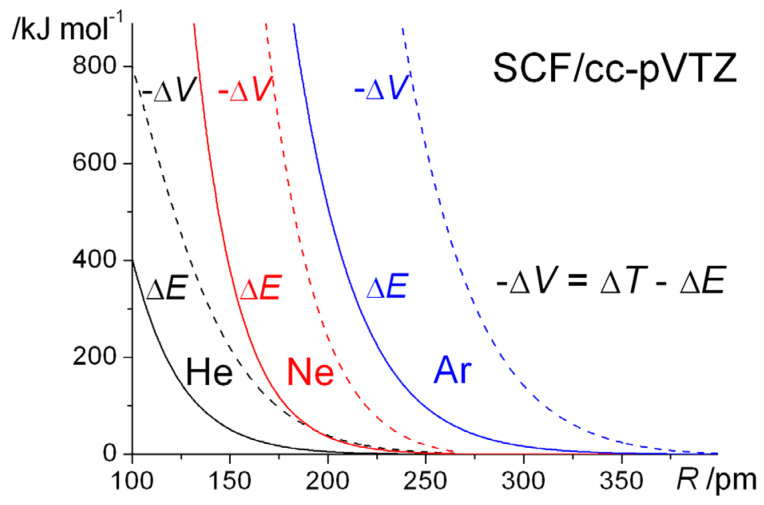
Helium (in black), neon (in red), and argon (in blue) dimers: total interaction energies (Δ*E*), shown in full lines, computed at the simple mean field (SCF) level of theory using the correlation consistent triple zeta basis sets. Comparison with the respective potential (−ΔV) and kinetic components (Δ*T*–Δ*E*) shown by dashed lines.

**Table 1 molecules-25-02667-t001:** H_2_^+^, H_2_: Computed molecular energies (*E*) and their kinetic (*T*) and potential (*V*) components (relative to H and H + H, respectively) at their equilibrium bond lengths (*R*_e_) with fixed and optimized orbital exponents.

	H_2_^+^		H_2_	
***ζ***	1.0	1.239	1.0	1.167
***R*_e_/*a*_0_**	2.49	2.00	1.64	1.41
**Δ*E*/*E*_h_**	−0.065	−0.087	−0.116	−0.139
**Δ*T*/*E*_h_**	−0.117	0.087	−0.162	0.139
**Δ*V*/*E*_h_**	0.052	−0.174	0.046	−0.278
**−*V*/*T***	2.48	2.00	2.33	2.00

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
