# Peer review of "The Basics of Covalent Bonding in Terms of Energy and Dynamics"

_molecules, 2020, doi:10.3390/molecules25112667_

Round 1

Reviewer 1 Report

Here are the comments and suggestions concerning the title manuscript. 

Firstly, the title and the style of the main body of the text are more likely to be the chapter of an educational book than a modern review. Maybe the authors can briefly comment on what is the significant improvement of the data beyond described previously in "Covalent Bonding in the Hydrogen Molecule" in J. Phys. Chem. A and what other points of view and new approaches authors discuss here in comparison to their previously suggested picture of covalent bonding in H2

Secondly, the reference list consists of 95 items, while only approximately 1/4 of them are published after the 2000 year and 1/3 of those are published by the authors. So I think the coverage of modern works is not enough although I understand that mainly this is because of the ancientry of the theme in the field of covalent bonding. But still insufficiency of modern data leads to a lack of comparison of different sources in the main body of the Review. 

All the above-written does not influence the fact that the manuscript is well-written in good scientific style, has a consistent composition and clear English. 

Reviewer 2 Report

I greatly enjoyed reading this beautiful review by the authors who give a very sound and detailed analysis of the physical mechanism which leads to a chemical bond in the archetypical molecules H2 and H2+. I actually reviewed a previous original paper by these authors some years ago where they discussed and presented the interpretation of the various factors which lead to a chemical bond in the two molecules. The present work is a summary of their study, which is based on the original work by Ruedenberg and was further elaborated by W. H. E. Schwarz and the present authors. The writing is clear and I hope that the paper contributes to the acceptance of the relevance of kinetic energy for chemical bond formation. I only have one comment which might be considered by the authors. I would be great, if they perhaps analyze with the same in-depth approach the chemical bond in the heavier species N2, which is a good example for showing that the exchange (Pauli) repulsion is an additional important part of chemical bonding for molecules, which possess more than two electrons. I recommend publication of this beautiful work as it is.

Round 2

Reviewer 1 Report

Dear Authors and Editors,

I agree that both educational and philosophical types of reviews can be interesting to the audience, so I hope that this Manuscript finds its readers.